# What controls ozone sensitivity in the upper tropical troposphere?

Clara M. Nussbaumer[1], Horst Fischer[1], Jos Lelieveld[1,2], and Andrea Pozzer[1,2]

[1]Max Planck Institute for Chemistry, Department of Atmospheric Chemistry, Mainz, Germany
[2]Climate and Atmosphere Research Center, The Cyprus Institute, Nicosia, Cyprus

**Correspondence:** Clara Nussbaumer (clara.nussbaumer@mpic.de)

**Abstract.**

Ozone is an important contributor to the radiative energy budget of the upper troposphere (UT). Therefore, observing and understanding the processes contributing to ozone production are important for monitoring the progression of climate change. Nitrogen oxides ($NO_x \equiv NO + NO_2$) and volatile organic compounds (VOC) are two main tropospheric precursors to ozone formation. Depending on their abundances, ozone production can be sensitive to changes in either of these two precursors. Here, we focus on processes contributing to ozone chemistry in the upper tropical troposphere between $30\,°$S and $30\,°$N latitude, where changes in ozone have a relatively large impact on anthropogenic radiative forcing. Based on modeled trace gas mixing ratios and meteorological parameters simulated by the EMAC atmospheric chemistry - general circulation model, we analyze a variety of commonly applied metrics including ozone production rates ($P(O_3)$), the formaldehyde (HCHO) to $NO_2$ ratio and the share of methyl peroxyradicals ($CH_3O_2$) forming HCHO ($\alpha(CH_3O_2)$), for their ability to describe the chemical regime. We show that the distribution of trace gases in the tropical UT is strongly influenced by the varying locations of deep convection throughout the year, and we observe peak values for $NO_x$ and $P(O_3)$ over the continental areas of South America and Africa where lightning is frequent. We find that $P(O_3)$ and its response to NO is unsuitable for determining the dominant regime in the upper troposphere. Instead, $\alpha(CH_3O_2)$ and the HCHO/$NO_2$ ratio in combination with ambient NO levels perform well as metrics to indicate whether $NO_x$ or VOC sensitivity is prevalent. We show that effectively only the knowledge of the availability of NO and $HO_2$ is required to adequately represent $O_3$ precursors and its sensitivity towards them. A sensitivity study with halving, doubling and excluding lightning $NO_x$ demonstrates that lightning and its distribution in the tropics are the major determinants of the chemical regimes and ozone formation in the upper tropical troposphere.

## 1 Introduction

Ozone ($O_3$) is abundant in the stratosphere and makes life on earth possible by absorbing highly energetic UV radiation emitted by the sun (Rowland, 1991; Staehelin et al., 2001). In the troposphere, on the other hand, high $O_3$ levels have adverse effects on human health, plant growth and climate (Ainsworth et al., 2012; Cooper et al., 2014; Nuvolone et al., 2018). Ground-level tropospheric ozone has received particular attention due to its role in causing cardiovascular and respiratory diseases (Nuvolone et al., 2018). Additionally, ozone can be detrimental to plants through limiting stomatal conductance and therefore the capability of to perform plants photosynthesis (Ainsworth et al., 2012; Mills et al., 2018). Ozone in the free troposphere is subject to particular focus due to its radiative forcing efficiency as a greenhouse gas and its contribution to global warming

and climate change. Ozone is the third most important anthropogenic greenhouse gas after carbon dioxide ($CO_2$) and methane ($CH_4$), with a particularly strong impact in the upper troposphere where concentrations of the natural greenhouse gas water vapor are small compared to the surface. Changes in ozone exert (and will continue to exert) a particularly large impact on

the earth's radiative forcing – especially in the tropopause region and the tropical UT (Lacis et al., 1990; Mohnen et al., 1993; Wuebbles, 1995; Lelieveld and van Dorland, 1995; van Dorland et al., 1997; Staehelin et al., 2001; Iglesias-Suarez et al., 2018; Skeie et al., 2020).

While transport from the stratosphere contributes significantly to ozone in the upper troposphere, the formation of $O_3$ from its precursors nitrogen oxides ($NO_x$) and volatile organic compounds (VOCs), might still be the predominant source of ozone in

this layer of the atmosphere (Lelieveld and Dentener, 2000; Cooper et al., 2014; Pusede et al., 2015). In the lower troposphere, $NO_x$ mostly originates from combustion processes such as vehicle engines and industrial activity. Soil emissions, partly natural and partly from agricultural activity, additionally contribute to $NO_x$ sources at the surface. In the upper troposphere, $NO_x$ is derived from lightning and aircraft (Pusede et al., 2015). VOC sources are even more diverse and range from biogenic vegetation emissions to anthropogenic emissions like combustion processes or volatile chemical products, such as paints,

detergents, cosmetics (McDonald et al., 2018). Previous studies have shown that methane ($CH_4$) is one of the most important VOC precursors to $O_3$ in the upper troposphere (Moxim and Levy, 2000; Cooper et al., 2006). Within a photochemical cycle catalyzed by OH radicals, VOCs and nitric oxide (NO) molecules form nitrogen dioxide ($NO_2$), which can subsequently react with $O_3$ in the presence of oxygen and sunlight as shown in the overall Reaction (R1) (Leighton, 1961; Crutzen, 1988).

$$NO_2 + O_2 \xrightarrow{h\nu} NO + O_3 \qquad\qquad\qquad (R1)$$

Deviations from the $HO_x$ cycle, including self-reactions of peroxy radicals, (Reactions (R2)), and the reaction of OH with $NO_2$ forming $HNO_3$ (Reaction (R3)), can terminate the formation of ozone. A detailed description of the $HO_x$ cycle and its termination reactions can, for example, be found in Pusede and Cohen (2012), Pusede et al. (2015) and Nussbaumer and Cohen (2020).

$$HO_2 + HO_2 \rightarrow H_2O_2 + O_2 \qquad RO_2 + HO_2 \rightarrow ROOH + O_2 \qquad RO_2 + RO_2 \rightarrow ROOR + O_2 \qquad (R2)$$


$$NO_2 + OH \rightarrow HNO_3 \qquad\qquad\qquad (R3)$$

Depending on the availability of its precursors, ozone formation can either be sensitive to the levels of $NO_x$ or VOC. While terms like $NO_x$ or VOC -"limited" -"sensitive" and -"saturated" are widely used in the literature in reference to chemical ozone regimes, there is no unified definition, as pointed out in a review by Sillman (1999) more than two decades ago. These

metrics are well studied and reported at surface levels. However, most of the indicators for either regime are no longer valid when it comes to the upper troposphere. This is due, for example, to the changing ratios of the investigated trace gases with

altitude and the decreasing temperature, which affects chemical rate constants and, therefore, the relevance of specific reactions (e.g. $NO_2 + OH$). Due to its wide-ranging atmospheric effects, not only on air quality and human health at the surface but also on atmospheric oxidation processes (the self-cleaning mechanism) and on climate change, with the upper troposphere being especially sensitive, it is highly relevant to understand and monitor $O_3$ levels and the main drivers of its sensitivity also at these altitudes.

Initial descriptions of ozone chemistry and the coining of the term "regime" date back to the late 1980s with studies by Liu et al. (1987), Lin et al. (1988) and Sillman et al. (1990). The most common definition for chemical regimes in the literature is based on the response of ozone production ($P(O_3)$) to changes in its precursors based on the ozone isopleths, which is described in review articles and textbooks (National Research Council, 1992; Seinfeld and Pandis, 1998; Sillman, 1999; Seinfeld, 2004). Correspondingly, in low-$NO_x$ environments increases in $NO_x$ lead to increases in $O_3$, while changes in VOCs have little to no impact – a $NO_x$-sensitive regime. In high-$NO_x$ environments increases in $NO_x$ affect decreases in $O_3$ – a VOC-sensitive regime (or $NO_x$-saturated regime). Within an $NO_x$-sensitive regime, OH radicals primarily react with VOCs and promote the catalytic $HO_x$ cycle and the formation of $O_3$. The self-reaction of peroxy radicals (R2) is the main termination reaction. With increasing $NO_x$ levels and the transition to a VOC-sensitive regime, the termination reaction of OH with $NO_2$ to form $HNO_3$ (R3) becomes dominant, affecting the anti-proportional correlation of $NO_x$ and $O_3$.

Various indicators have been reported in the literature to determine the dominant regime. Table 1 provides an overview of the most important metrics found in literature and the following paragraph presents and discusses these and some more indicators in detail.

Some studies have directly addressed the production of $O_3$ (or odd oxygen ($O_x$) $\equiv O_3 + NO_2$) in response to changing $NO_x$ (Brasseur et al., 1996; Jaeglé et al., 1999; Tonnesen and Dennis, 2000a; Tadic et al., 2021). Other studies considered the so-called ozone production efficiency (OPE), which evaluates how many ozone molecules are formed by $NO_x$ before it is removed to reaction products such as $HNO_3$ or PAN (peroxyacetyl nitrate) (Liu et al., 1987; Trainer et al., 1993; Wang et al., 2018a). Low OPEs indicate a VOC-sensitive and high OPEs a $NO_x$-sensitive regime. Similar approaches such as the ratio of $O_3$ and reactive nitrogen species ($NO_y$) or $NO_z$ ($\equiv NO_y - NO_x$) have also been reported (Milford et al., 1994; Sillman, 1995; Fischer et al., 2003; Peralta et al., 2021; Wang et al., 2022b).

A common method for determining the dominant regime in urban environments is the weekend ozone effect, where the response of $O_3$ levels to decreasing $NO_x$ mixing ratios on weekends is monitored (e.g., Fujita et al. (2003); Pusede and Cohen (2012); Nussbaumer and Cohen (2020); Sicard et al. (2020); Gough and Anderson (2022)).

Another indicator is the ratio between formaldeyhde (HCHO) and $NO_2$. Sillman (1995) originally suggested the ratio $HCHO/NO_y$ ($NO_y \equiv NO_x + HNO_3 +$ organic nitrates) as a metric, which was later adjusted to the $HCHO/NO_2$ ratio. This metric evaluates the reaction of OH radicals with VOCs (ultimately leading to HCHO as a reaction intermediate) enhancing $O_3$ production in competition with the reaction of OH radicals with $NO_2$, which decelerates $O_3$ formation (Tonnesen and Dennis, 2000b). The $HCHO/NO_2$ ratio has been widely applied in the literature based on ground-based measurements and satellite observations (e.g., Duncan et al. (2010); Jin et al. (2020); Xue et al. (2022)).

**Table 1.** Overview of the most common metrics to determine if $O_3$ chemistry is sensitive towards $NO_x$ or VOC reported in literature. Details can be found in the text.

| metric | required parameters | $NO_x$ sensitivity | VOC sensitivity | references | suitability |
|---|---|---|---|---|---|
| $P(O_3)$ | $NO_x$, $O_3$, $HO_2$, $RO_2$ | $P(O_3)$ increases with $NO_x$ | $P(O_3)$ decreases with $NO_x$ | Brasseur et al. (1996), Jaeglé et al. (1999), Tonnesen and Dennis (2000a), Tadic et al. (2021) | surface |
| OPE | $NO_x$ & reaction products (e.g. $HNO_3$, PAN) | high OPEs | low OPEs | Liu et al. (1987), Trainer et al. (1993), Wang et al. (2018a) | surface |
| weekend $O_3$ effect | $NO_x$, $O_3$ | $O_3$ decreases on weekends | $O_3$ increases on weekends | Fujita et al. (2003), Pusede and Cohen (2012), Nussbaumer and Cohen (2020), Sicard et al. (2020), Gough and Anderson (2022) | surface |
| HCHO to $NO_2$ ratio | HCHO, $NO_2$ | high $HCHO/NO_2$ e.g. $HCHO/NO_2 > 2$* | low $HCHO/NO_2$ e.g. $HCHO/NO_2 < 1$* | Sillman (1995), *Duncan et al. (2010), Jin et al. (2020), Xue et al. (2022) | surface |
| $H_2O_2$ to $HNO_3$ ratio | $H_2O_2$, $HNO_3$ | high $H_2O_2/HNO_3$ e.g. $H_2O_2/HNO_3 > 0.4$** | low $H_2O_2/HNO_3$ e.g. $H_2O_2/HNO_3 < 0.4$** | **Sillman (1995), Wang et al. (2018b), Vermeuel et al. (2019), Liu et al. (2021), Gough and Anderson (2022) | surface |
| $\alpha(CH_3O_2)$ | NO, OH, $HO_2$ | $\alpha(CH_3O_2)$ increases with NO | $\alpha(CH_3O_2)$ unaffected by NO | Nussbaumer et al. (2021c), Nussbaumer et al. (2022) | entire troposphere |

The ratio of hydrogen peroxide ($H_2O_2$) to $HNO_3$ is another metric used for regime analysis. Also initially suggested by Sillman (1995), it compares the $HO_2$ self-reaction (forming $H_2O_2$) with the reaction of OH and $NO_2$, both leading to termination of the $HO_x$ cycle. While the $HO_2$ self-reaction dominates over the formation of $HNO_3$ as a termination reaction, $O_3$ increases linearly with $NO_x$. Recent studies using $H_2O_2/HNO_3$ include Wang et al. (2018b), Vermeuel et al. (2019) and Liu et al. (2021). The

$HCHO/NO_2$ and $H_2O_2/HNO_3$ ratios, as well as the OPE require absolute values as reference points to determine the regime, which can vary depending on the ambient conditions; background mixing ratios are a major drawback of these metrics.

Dyson et al. (2022) recently analyzed the dominant regime in Beijing using a method that considers the loss of OH, $HO_2$ and $RO_2$ radicals via reaction with $NO_x$ in comparison to the overall production of these radical species. The production thereby equals the overall radical loss via reaction with $NO_x$, self-reaction and aerosol uptake, an idea which has been previously

described by Sakamoto et al. (2019). Within a VOC-sensitive regime, $HO_2$ is predominantly lost via the reaction with NO, while in a $NO_x$-sensitive regime, aerosol uptake plays a significant role in $HO_2$ loss. Dyson et al. (2022) found the transition to occur around 0.1 ppbv of NO.

Cazorla and Brune (2009) and Hao et al. (2023) reported direct measurements of $P(O_3)$ in a reaction chamber through observing changes in $O_x$ in a certain time interval. This technique can be used to determine the dominant chemical regime when correlated

with ambient NO mixing ratios.

These metrics were developed for the conditions near the surface, and most of them have not been applied to higher altitudes in the troposphere. Of the above metrics, the HCHO to $NO_2$ ratio is the only one, which we find to be applicable also to high altitudes – but only in combination with ambient NO mixing ratios. We demonstrate this in Section 3.3. So far, absolute thresholds (e.g. lower than 1 for VOC sensitivity or higher than 2 for $NO_x$ sensitivity (Duncan et al., 2010)) have been reported

and applied in the literature. Due to the strong vertical trace gas gradients, these absolute thresholds are not applicable in the upper troposphere.

Some studies (including Jaeglé et al. (1998), Wennberg et al. (1998) and Jaeglé et al. (1999)) have analyzed the dominant chemical regime in the UT. These studies focus on the U.S. and the North Atlantic and consistently report a linear correlation between $P(O_3)$ and $NO_x$ based on aircraft observations, deducing a $NO_x$-sensitive regime, while model simulations predict

a $P(O_3)$ decrease with high $NO_x$. One explanation for these observations could be that these studies, published around 25 years ago, overestimated the $NO_x$ loss. We know today that the reaction rate of $NO_2$ and OH is much lower than previously assumed (Mollner et al., 2010; Henderson et al., 2012; Nault et al., 2016). The loss reaction of $NO_2$ with OH to $HNO_3$ does not play a significant role under the conditions in the upper troposphere (in contrast to low-altitude) so that the typical definition for a VOC-sensitive regime where $O_3$ production decreases with increasing $NO_x$ does not apply anymore. Khodayari et al.

(2018) reported an $NO_x$-saturated (VOC-sensitive) regime based on a modeling sensitivity study, where globally a decreasing $O_3$ burden was observed with increasing lightning $NO_x$. We suggest that this observed anti-correlation might not result from increased $NO_x$ loss as applicable for surface conditions, and might instead be an outcome of decreasing $HO_2$ with increasing NO. Pickering et al. (1990) reported a VOC-sensitive regime over the U.S. at 11 km altitude based on measurements in June 1985 and model simulations. A study by Dahlmann et al. (2011) indicates increasing $P(O_3)$ with increasing NO at 250 hPa

over Europe, implying a $NO_x$-sensitive regime following the common definition. Shah et al. (2023) analyzed the relationship

between the $NO_y/NO$ ratio and $O_3$ mixing ratios and assumed an $NO_x$-sensitive regime over the Central U.S. based on a flight during the DC3 research campaign in 2012. Liang et al. (2011) analyzed changes in net ozone production with $NO_x$ in the Arctic troposphere and found a proportional relationship up to 10 ppbv $NO_x$ based on box model calculations and observations.

While all studies have briefly touched upon the dominant chemical regime in the upper troposphere, a thorough analysis and a definition that is valid throughout the troposphere have not yet been reported. In view of ozone's major implications for the earth's radiative energy budget and climate change (particularly in the UT), $O_3$ sensitivity is highly relevant for understanding and monitoring which precursors and processes are most important for the $O_3$ budget at high altitudes in the troposphere.

In Nussbaumer et al. (2021a), we introduced a new metric $\alpha(CH_3O_2)$ for determining the dominant regime, which presents the ratio of methyl peroxyradicals ($CH_3O_2$) forming HCHO with NO versus the reaction of $CH_3O_2$ with $HO_2$. We have applied this metric to ground-based observations at three different sites in Europe and for aircraft observations during the 2022 BLUESKY research campaign in the upper troposphere over Europe (Nussbaumer et al., 2022). We found a change at high altitudes from a VOC- to a $NO_x$-sensitive regime over the past two decades up to 2020, promoted by emission reductions during the COVID-19 pandemic.

In this study, we use $\alpha(CH_3O_2)$ to analyze the dominant regime in the upper tropical troposphere between $30\,^\circ$ S and $30\,^\circ$ N latitude based on modeled trace-gas mixing ratios and meteorological parameters by the EMAC atmospheric chemistry - general circulation model. We additionally investigate the effects of $NO_x$ produced by lightning in six different tropical areas: the Pacific Ocean, South America, the Atlantic Ocean, Africa, the Indian Ocean and South East Asia. Finally, we provide a new definition for $NO_x$- and VOC-sensitive regimes, which is valid throughout the troposphere.

## 2 Methods

### 2.1 Calculations of ozone production (P($O_3$)) and loss (L($O_3$)) rates

The calculation of ozone production (P($O_3$)) and loss (L($O_3$)) rates was performed as presented in Section 2.1 of Nussbaumer et al. (2022). Briefly, ozone production P($O_3$) is described by the reaction of NO with $HO_2$ and peroxy radicals $R_zO_2$ (Equation (1)); the latter can be approximated by $CH_3O_2$ in the upper troposphere. $CH_3O_2$ accounts for $85 \pm 5\,\%$ of $R_zO_2$, represented by the sum of $CH_3O_2$, $C_2H_5O_2$ (ethylperoxy radicals), $CH_3CO_3$ (peroxyacetyl radicals), $CH_3COCH_2O_2$ (acetonylperoxy radicals), iso-$C_3H_7O_2$ (iso-propylperoxy radicals), $C_5H_6O_3$ (isoprene (hydroxy) peroxy radicals), $C_4H_7O_4$ (methyl vinyl ketone / methacrolein peroxy radicals) and LHOC$_3$H$_6$O$_2$ (hydroxyperoxy radicals from propene + OH).

$$P(O_3) = k_{NO+HO_2} \times [HO_2] \times [NO] + \sum_z k_{NO+R_zO_2} \times [R_zO_2][NO]) \tag{1}$$

Ozone loss L($O_3$) is calculated as shown in Equation (2) via the reaction of $O_3$ with $HO_2$ and OH and via photolysis. The latter only yields an effective ozone loss if O($^1$D) (resulting from $O_3$ photolytic cleavage) reacts with $H_2O$ instead of colliding with $O_2$ or $N_2$ (and reforming $O_3$). This share is represented by $\alpha_{O(^1D)}$ in Equation (3).

$$L(O_3) = k_{O_3+HO_2} \times [HO_2] \times [O_3] + k_{O_3+OH} \times [OH] \times [O_3] + \alpha_{O^1D} \times j(O^1D) \times [O_3] \qquad (2)$$

$$\alpha_{O^1D} = \frac{k_{O^1D+H_2O} \times [H_2O]}{k_{O^1D+N_2} \times [N_2] + k_{O^1D+O_2} \times [O_2] + k_{O^1D+H_2O} \times [H_2O]} \qquad (3)$$

The resulting net ozone production rate (NOPR) is then calculated by subtracting ozone loss from its production as shown in Equation (4).

$$\begin{aligned} NOPR = P(O_3) - L(O_3) \\ = [NO] \times (k_{NO+HO_2} \times [HO_2] + k_{NO+CH_3O_2} \times [CH_3O_2]) \\ - [O_3] \times (k_{O_3+HO_2} \times [HO_2] + k_{O_3+OH} \times [OH] + \alpha_{O^1D} \times j(O^1D)) \end{aligned} \qquad (4)$$

## 2.2 Calculations of $\alpha(CH_3O_2)$

$\alpha(CH_3O_2)$ represents the share of methyl peroxyradicals forming HCHO with NO and OH versus the reaction with $HO_2$ yielding $CH_3OOH$ and is calculated as shown in Equation (5).

$$\alpha_{CH_3O_2} = \frac{k_{CH_3O_2+NO} \times [NO] + k_{CH_3O_2+OH} \times [OH]}{k_{CH_3O_2+NO} \times [NO] + k_{CH_3O_2+OH} \times [OH] + k_{CH_3O_2+HO_2} \times [HO_2]} \qquad (5)$$

We demonstrated in previous studies that $\alpha(CH_3O_2)$ can be used as a metric to determine the dominant chemical regime (Nussbaumer et al., 2021a, 2022). The formation of HCHO from $CH_3O_2$ through reaction with NO leads to $O_3$ formation, as $NO_2$ is formed simultaneously, which can subsequently react to $O_3$ via photolysis. In contrast, the reaction of $CH_3O_2$ with $HO_2$ represents a termination reaction of the $HO_x$ cycle and therefore decelerates $P(O_3)$. $CH_3O_2$ here is a proxy for VOCs which form HCHO and $O_3$ through $RO_2$. A principal precursor to $CH_3O_2$ is $CH_4$, which is oxidized by OH radicals and reacts with $O_2$ in the first two steps of the catalytic $HO_x$ cycle. The atmospheric $CH_4$ concentration is increasing rapidly, with the tropics implicated, suggestive of feedbacks that may accelerate $CH_4$ growth in future contributing to $O_3$ formation in the upper troposphere (Griffiths et al., 2021; Nisbet et al., 2023). Other precursors to $CH_3O_2$ can be acetone, methyl hydroxy peroxide or acetaldehyde through photolysis or reaction with OH radicals (Nussbaumer et al., 2021a). The advantage of considering $CH_3O_2$ is that it represents an entire group of VOCs that can form $O_3$, including $CH_4$, instead of handling multiple trace gases and risking incompleteness. The progression of $\alpha(CH_3O_2)$ in dependence of the ambient NO mixing ratio is shown in Figure 1. The black line presents the average $\alpha(CH_3O_2)$ across all longitudes and between $30\,°$ S and $30\,°$ N latitude at $200\,hPa$ altitude for daily values from 2000 to 2019 binned to the NO mixing ratios, as obtained from the EMAC modeling output. It therefore describes the background behavior of NO vs $\alpha(CH_3O_2)$ for all data used in this study. The grey error shades show the $1\,\sigma$ standard deviation resulting from the averaging. At low NO mixing ratios (here $< 0.1$ ppbv), $\alpha(CH_3O_2)$ changes rapidly even

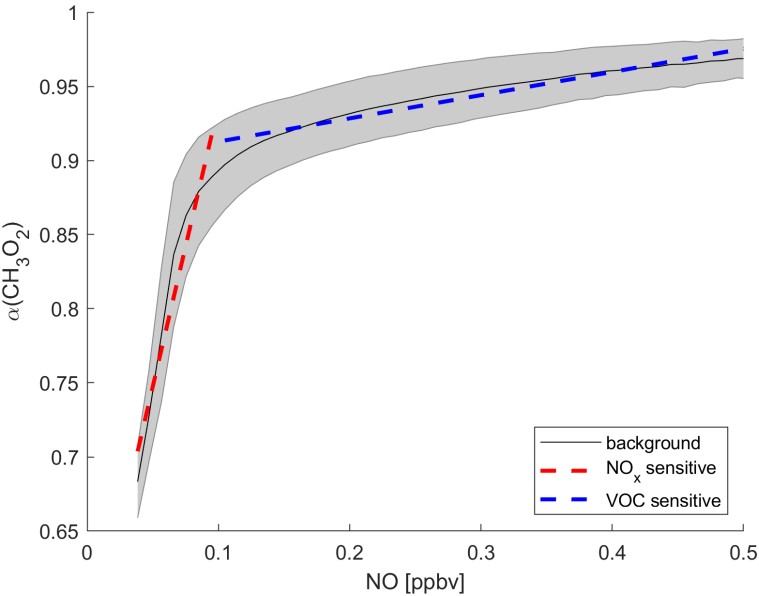

**Figure 1.** Illustration of how $\alpha(CH_3O_2)$ can serve as a metric to determine the dominant chemical regime. The black line shows the tropical background $\alpha(CH_3O_2)$ binned to NO mixing ratios from the EMAC model output. The grey shading shows the $1\sigma$ standard deviation. The red dashed line is a linear fit for NO < 0.1 ppbv and represents $NO_x$-sensitive $O_3$ chemistry. The blue dashed line represents VOC-sensitive $O_3$ chemistry for NO > 0.1 ppbv.

with small changes in NO. The resulting slope of the linear fit of the data is $3.75 \pm 0.44$ ppbv$^{-1}$. In this range, $CH_3O_2$ reacts both with NO and with $HO_2$ (and with itself). With increasing availability of NO, the reaction of $CH_3O_2$ with NO and therefore the amount of $O_3$ formed is enhanced. This regime is referred to as $NO_x$-sensitive. In comparison, for higher NO mixing ratios (here > 0.1 ppbv), $\alpha(CH_3O_2)$ only shows minor changes with increasing NO and is almost constant. The resulting slope is $0.16 \pm 0.01$ ppbv$^{-1}$. In this range, NO is so abundant that $CH_3O_2$ reacts primarily with NO and changes in NO have almost no impact on the reaction. The amount of $O_3$ formed is limited by the abundance of $CH_3O_2$, which itself is formed by a precursor VOC, and no longer increases with increasing NO. This regime is referred to as VOC-sensitive. Depending on where in this graph the data points from specific areas are located, it is possible to identify if a $NO_x$- or a VOC-sensitive regime is dominant. This method underlines that $O_3$ sensitivity is dependent on the availability of NO and $HO_2$ radicals. Individual VOCs do not need to be considered when investigating $O_3$ sensitivity, as $HO_2$ effectively represents their chemical impact.

## 2.3 Modeling study

The data analyzed in this study were produced by model simulations using the ECHAM5 (fifth generation European Centre Hamburg general circulation model, version 5.3.02)/MESSy2 (second-generation Modular Earth Submodel System, version 2.54.0) Atmospheric Chemistry (EMAC) model. Details on the EMAC model can be found in Jöckel et al. (2016). We applied EMAC in the T63L47MA-resolution, i.e., with a spherical truncation of T63 (corresponding to a quadratic Gaussian grid of

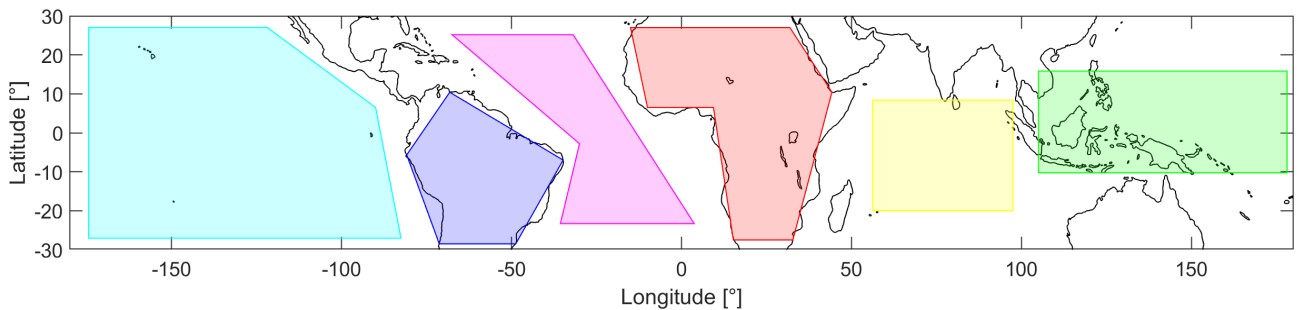

**Figure 2.** Overview of the defined areas in the tropics between $30\,^{\circ}$ S and $30\,^{\circ}$ N latitude: Pacific Ocean (cyan), South America (blue), Atlantic Ocean (pink), Africa (red), Indian Ocean (yellow) and South East Asia (green).

195  1.875 by 1.875 degrees in latitude and longitude) with 47 vertical hybrid pressure levels up to 0.01 hPa. Roughly 22 levels are included in the troposphere depending on the latitude, and the model has a time step of 6 minutes. The dynamics of the EMAC model have been weakly nudged in the troposphere (Jeuken et al., 1996) towards the ERA5 meteorological reanalysis data (Hersbach et al., 2020) of the European Centre for Medium-Range Weather Forecasts (ECMWF) to represent the actual day-to-day meteorology in the troposphere. The set-up adopted here is similar to the one presented in Reifenberg et al. (2022),

200  using the anthropogenic emissions CAMS-GLOB-ANTv4.2 (Granier et al., 2019), with varying monthly values for the period 2000–2019. The model has been extensively evaluated for ozone (e.g., Jöckel et al., 2016), showing a systematic though minor overestimation of the model compared to observations, which is a common feature in chemistry general circulation models of this complexity (Young et al., 2013). Comparison of the model results against numerous field campaigns (e.g., Lelieveld et al., 2018; Tadic et al., 2021; Nussbaumer et al., 2022) reveals a good agreement between observations and model results

205  of $NO_x$ and VOCs for locations in the UT. The reference simulation covers the time period 2000–2019 with hourly output of trace gas mixing ratios of $O_3$, $NO$, $NO_2$, $OH$, $HO_2$, $CH_3O_2$, $HCHO$, $CO$, $CH_4$ and $H_2O$, as well as the photolysis rates $j(NO_2)$ and $j(O^1D)$ and meteorological parameters such as temperature and pressure, necessary for calculating net ozone production rates and $\alpha(CH_3O_2)$. The data were post-processed to obtain daily values at local noon time and calculated for 200 hPa (upper troposphere) using bilinear interpolation between the hybrid pressure model levels. This pressure level was chosen to represent

210  upper tropospheric influence in the tropics in order to capture lightning events and avoid strong influence of transport from the stratosphere. The choice is also underlined by the results presented in Figure 7 where 200 hPa marks the area of transition from $NO_x$ sensitivity in the free troposphere to VOC sensitivity in the stratosphere, impacted by lightning.

In this work the emissions of $NO_x$ from lightning are following the work of Tost et al. (2007). A constant $NO_x$ emission of $\sim 20.6$ kg N/flash is used for cloud-to-ground flashes. The ratio of $NO_x$ production by intra-cloud flashes is lower by a factor of

215  0.1. The lightning frequency is estimated with the parameterization of Grewe et al. (2001). Here the updraft velocity (used as proxy for convective strength) is associated with cloud electrification and therefore proportional to frequency of the lightning flashes. On the other side, aircraft emissions are prescribed from the CAMS dataset (CAMS-GLOB-AIR v1.1, Granier et al., 2019).

For detailed analysis, six different areas are defined and their geographic extent is shown in Figure 2. These areas refer to the Pacific Ocean (cyan), South America (blue), the Atlantic Ocean (pink), Africa (red), the Indian Ocean (yellow) and South East Asia (green).

## 3 Results and Discussion

### 3.1 Development of trace gases over time

The analyzed trace gases do not show statistically significant trends over time from 2000 to 2019 at 200 hPa, which we show in Figure S1 of the Supplement. We find small global increases of some trace gases, e.g., average NO and $HO_2$ mixing ratios increase by $\sim 5\%$ and average $NO_2$ and $O_3$ mixing ratios by up to $10\%$ from 2000 to 2019. Global mean temperature increases by approximately $1\,°C$ over the 20 year period. Even though slight trends can be detected, the variability is high and the $1\,\sigma$ standard deviation (grey shaded) is significantly larger than the variation over time and we therefore used a daily climatology from the 20-year period in order to simplify the calculations.

### 3.2 Tropical distribution

#### 3.2.1 $NO_x$

Figure 3 shows the distribution of NO in the upper tropical troposphere (at 200 hPa). We find large changes throughout the year related to the seasonality of deep convection and the location of the intertropical convergence zone (ITCZ). Deep convection dominates in the southern hemisphere in January, and during July it is most prevalent in the northern hemisphere Yan (2005). In order to illustrate the differences, we subdivide the data into four periods, December–February (DJF), March–May (MAM), June–August (JJA) and September–November (SON). Each grid cell extends over $1.875\,° \times 1.875\,°$ latitude and longitude and represents a 20-year average of the respective period.

During DJF, NO mixing ratios are highest over the tropical continental areas of South America, southern Africa and northern Australia, with average peak values between 0.3 and 0.4 ppbv. In comparison, NO mixing ratios are much lower over the Pacific and the Indian Oceans ($0.09 \pm 0.01$ ppbv), the Atlantic Ocean ($0.12 \pm 0.02$ ppbv), North Africa ($0.10 \pm 0.03$ ppbv) and South East Asia ($0.08 \pm 0.01$ ppbv). Generally, the mixing ratios over land are much higher than those over the ocean, and the mixing ratios north of the equator ($0.09 \pm 0.03$ ppbv) are lower compared to south of the equator ($0.14 \pm 0.06$ ppbv). During MAM, NO mixing ratios over South America are similar to those during DJF ($0.21 \pm 0.05$ ppbv). NO mixing ratios over Africa are almost twice as high on average compared to DJF and reach peak values of 0.53 ppbv. The relative maxima relocate from South to Central Africa from DJF to MAM and also over the Arabian Peninsula and South Asia, including India. Mixing ratios over Australia are around 0.15 ppbv and approximately half of those during DJF and are similar over South East Asia. During JJA, peak NO mixing ratios are found north of the equator over Central America and North Africa. Average NO mixing ratios are almost $60\%$ higher in the northern ($0.14 \pm 0.07$ ppbv) compared to the southern tropical hemisphere ($0.09 \pm 0.02$ ppbv). The

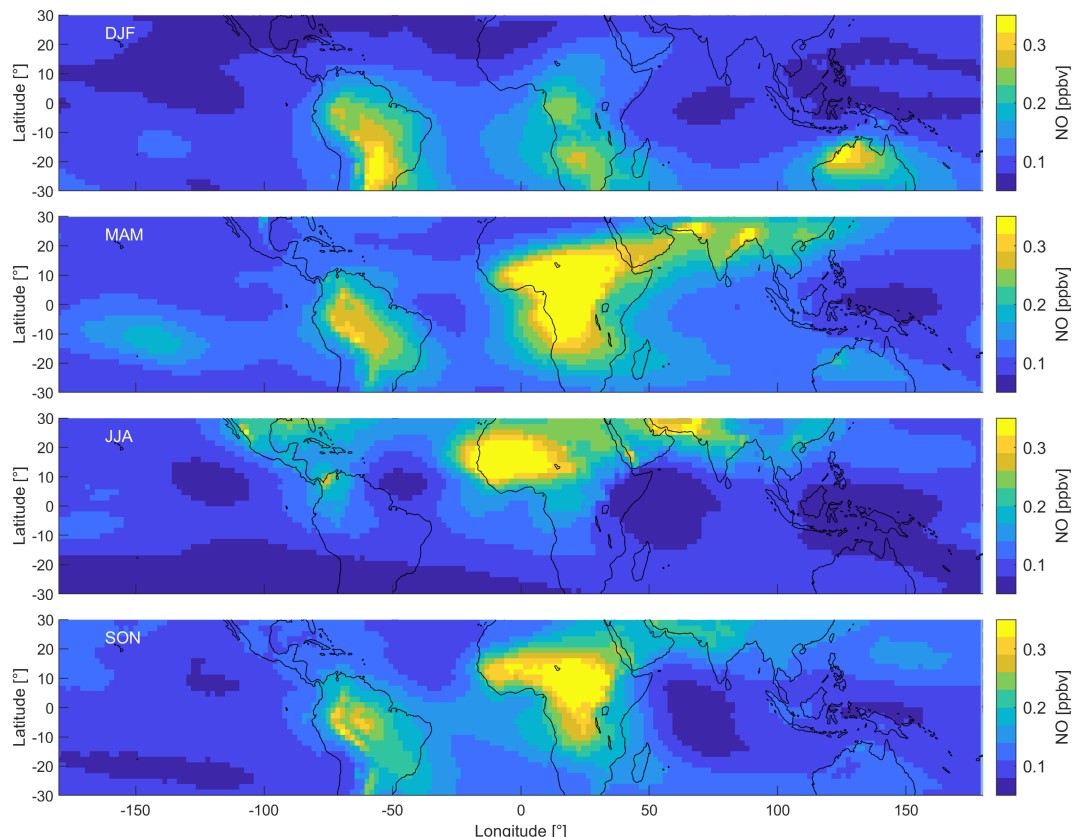

**Figure 3.** Distribution of NO in the tropical UT between 30 ° S and 30 ° N for December–February (DJF), March–May (MAM), June–August (JJA) and September–November (SON).

distribution, therefore, differs drastically from DJF. During SON, NO mixing ratios are similar to MAM and peak over South
America and Central Africa.

The highest NO mixing ratios are found in the locations of predominant deep convection, which vary throughout the year. In July, deep convection is highest over Central America, North Africa and South Asia (northern India). In January, it is predominant over South America, Central to South Africa and North Australia (Yan, 2005). The areas where these convective processes are prevailing define the ITCZ where north- and southeasterly trade winds converge. Increased thunderstorm activity
explains the occurrence of peak NO mixing ratios. Various studies have reported significantly increased lightning over land compared to the ocean, which is in line with the distribution of NO as shown in Figure 3 (Christian et al., 2003; Rudlosky and Virts, 2021; Nussbaumer et al., 2021c). South East Asia is often referred to as the "maritime" continent. This region experiences frequent cumulonimbus activity, but the convective available potential energy (CAPE) is less compared with that over the South American and in particular the African land masses. This region therefore shows lower NO mixing ratios throughout the year.

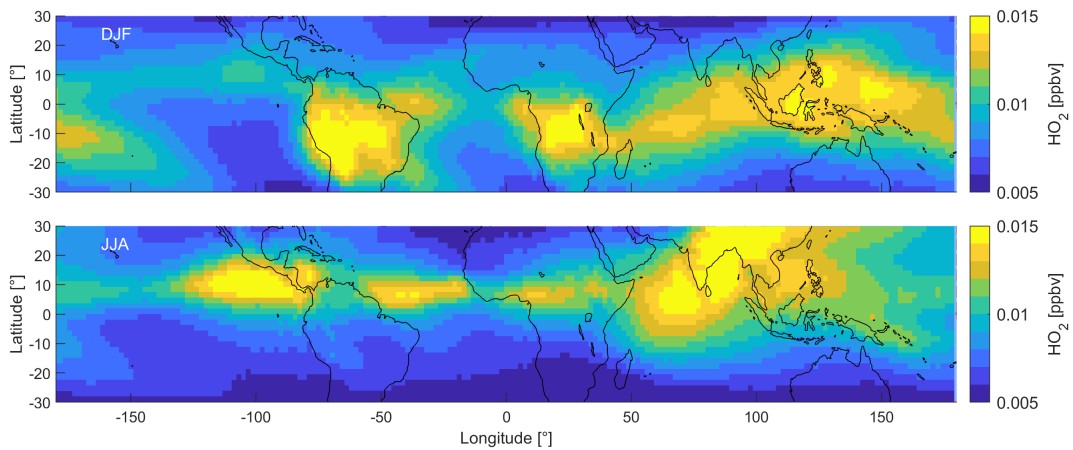

**Figure 4.** Distribution of HO$_2$ in the tropical UT between $30\,^\circ$ S and $30\,^\circ$ N during DJF (top panel) and JJA (bottom panel).

The relative distribution of NO$_2$ is very similar to NO, which we show in Figure S2 of the Supplement. On average, NO$_2$ mixing ratios are around a factor of 7 lower compared to NO.

### 3.2.2 HO$_2$

Figure 4 shows the DJF and JJA distributions of HO$_2$. An overview of all four periods can be found in Figure S3 of the Supplement. Similar to NO, the spatial DJF distributions of HO$_2$ mixing ratios show peak values between 15 and 20 pptv over

South America and South Africa. While NO shows minimum values over South East Asia and the Indian Ocean, HO$_2$ mixing ratios are elevated in these areas (12-13 pptv). Mixing ratios are lower over the Pacific and the Atlantic Oceans ($8\text{-}9 \pm 1$ pptv) and north of $\sim 20\,^\circ$ N. During JJA, HO$_2$ mixing ratios are elevated over the Indian Ocean, South Asia and Central America, including the Atlantic and Pacific Ocean around $10\,^\circ$ N latitude. HO$_2$ is relatively low over South America and Africa. During MAM and SON, HO$_2$ is mostly intermediate between DJF and JJA and does not show any noteworthy features. Mixing ratios

of CH$_3$O$_2$ show a very similar distribution to HO$_2$ across the tropical UT and range from $\sim 0.5$ to 4 pptv, which we show in Figure S4 of the Supplement.

### 3.2.3 NOPR

The tropical UT distribution of net ozone production rates is closely related to the distribution of NO (Figure 3). We show the model-calculated results for each period in Figure S5 of the Supplement. During DJF, NOPRs peak over South Amer-

ica with average values of $0.77 \pm 0.20$ ppbv h$^{-1}$ and maximum values above 1 ppbv h$^{-1}$ over southern Africa and northern Australia. In the course of the year and the local variations in deep convection, NOPR peaks move northwards, reaching the northernmost point in JJA, and moving southwards again in SON and DJF. During DJF, NOPRs are almost 60 % higher in the southern ($0.36 \pm 0.21$ ppbv h$^{-1}$) compared to the northern tropical hemisphere ($0.23 \pm 0.09$ ppbv h$^{-1}$). In contrast during JJA, NOPRs are more than twice as high in the northern ($0.36 \pm 0.15$ ppbv h$^{-1}$) compared to the southern tropical hemisphere

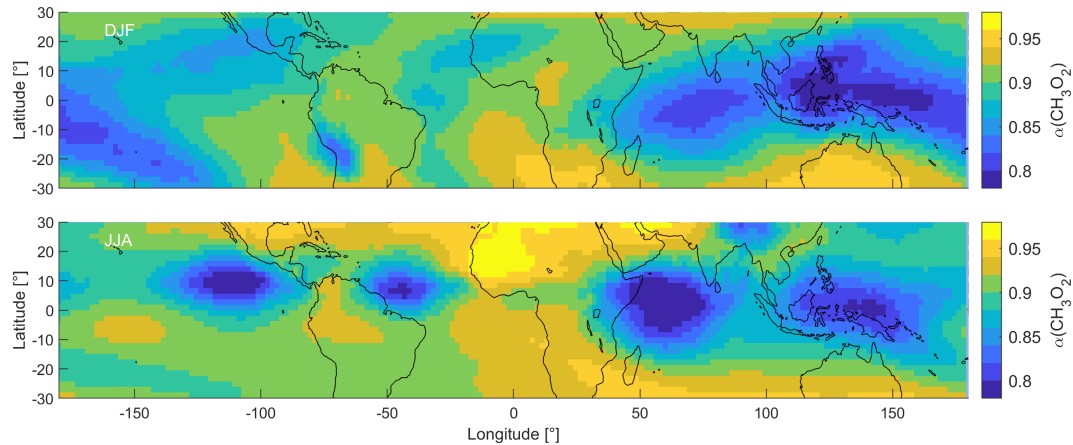

**Figure 5.** Distribution of $\alpha(CH_3O_2)$ in the tropical UT between $30\,^{\circ}$ S and $30\,^{\circ}$ N during DJF (top panel) and JJA (bottom panel).

$(0.17 \pm 0.06\,\mathrm{ppbv\,h^{-1}})$. The production of $O_3$ outweighs its loss by a factor of 8 on average for the studied conditions. The difference is larger in regions with peak NOPRs, e.g., over South America with a factor of 11, and smaller in regions with low NOPRs, e.g., over the Pacific Ocean with a factor of around 7. We show the distribution of both $P(O_3)$ and $L(O_3)$ in Figures S6 and S7 of the Supplement. These results, showing similar features for NOPRs and NO geographically, are in line with findings by Apel et al. (2015), who reported enhanced ozone production for high lightning $NO_x$ over the U.S. during a research flight

in June 2012 as part of the DC3 campaign.

### 3.2.4   $\alpha(CH_3O_2)$

Figure 5 shows the distribution of $\alpha(CH_3O_2)$ in the tropical UT during DJF and JJA. We show all periods in Figure S8 of the Supplement. During DJF, $\alpha(CH_3O_2)$ ranges from 0.77 to 0.95 with lowest values over South East Asia and highest values over South Africa and Australia. During JJA, lowest values are obtained over South East Asia, the Indian Ocean and over the Pacific

and Atlantic Oceans around $10\,^{\circ}$ N latitude. Maximum values of up to 0.97 are reached over North Africa and the Arabian Peninsula. Therefore, as expected, $\alpha(CH_3O_2)$ is proportional to NOPR and $NO_x$ mixing ratios and is anti-proportional to $HO_2$ mixing ratios. At low $NO_x/HO_2$ ratios, increases in NO enhance $\alpha(CH_3O_2)$, while at high $NO_x/HO_2$ ratios, changes in NO have no or only little effect. We will discuss the implications of $\alpha(CH_3O_2)$ for the dominant chemical regime in the tropical UT and specific regions in the following section.

### 295  3.3   Chemical regimes

#### 3.3.1   Baseline scenario

Figure 6 presents $\alpha(CH_3O_2)$, $O_3$ and the HCHO/$NO_2$ ratio binned to NO mixing ratios during DJF, MAM, JJA and SON. The graphs show NO mixing ratios up to 0.5 ppbv, which includes 99.6 % of all data points. The frequency distribution of the NO

data can be seen in Figure S9 of the Supplement. The black lines and the grey shades represent the average of all data points

binned to NO, which we refer to as background, and the associated $1\,\sigma$ standard deviation. The colored data points show the average of the individual areas as shown in Figure 2. The error bars represent the $1\,\sigma$ variability. Data for the Pacific Ocean are shown in cyan, for South America in blue, for the Atlantic Ocean in magenta, for Africa in red, for the Indian Ocean in yellow and for South East Asia in green. In this section, we first discuss the background curves (representing an average of all data point), and we subsequently present our findings for the individual areas.

Background $\alpha(CH_3O_2)$ (left column: (a), (d), (g) and (j)) increases strongly with NO for mixing ratios below 0.1 ppbv with a slope of $3.75 \pm 0.44\,\mathrm{ppbv}^{-1}$. For example, an average increase of $\alpha(CH_3O_2)$ by 0.1 results from an increase of ambient NO by around 27 pptv. This characterizes the $NO_x$-sensitive regime. In contrast, for NO mixing ratios higher than 0.1 ppbv NO, increasing NO has only a minor effect on $\alpha(CH_3O_2)$ (slope $= 0.16 \pm 0.01\,\mathrm{ppbv}^{-1}$), which represents the VOC-sensitive regime. To reach an increase of $\alpha(CH_3O_2)$ by 0.1, ambient NO needs to increase by 625 pptv, a factor of $>20$ higher compared to the

low-$NO_x$ regime. Within the $NO_x$-sensitive regime, predominantly $CH_3O_2$ reacts with NO, forming $O_3$, as well as with $HO_2$, which does not result in formation of $O_3$. With increasing NO, the share of the reaction with NO (compared to the reaction with $HO_2$) increases, which in turn enhances $O_3$. In contrast, within the VOC-sensitive regime $CH_3O_2$ radicals mostly react with NO in any case and increases in NO do not affect $O_3$. This is illustrated in the middle column of Figure 6 ((b), (e), (h) and (k)): $O_3$ increases with NO for low NO mixing ratios and reaches a plateau for high NO mixing ratios. While the shift from the

$NO_x$- to the VOC-sensitive regime is relatively sharp for $\alpha(CH_3O_2)$, the transition for $O_3$ is broader and more difficult to relate to a NO mixing ratio. This graph is illustrative, but should not be used solely for determining the dominant chemical regime. In the right column (Figure 6 (c), (f), (i) and (l)), we present the $HCHO/NO_2$ ratio binned to NO mixing ratios. In the literature, mostly absolute values for the $HCHO/NO_2$ ratio are used to determine the chemical regime, meaning the ratio is calculated and compared to a certain threshold, for example $HCHO/NO_2 > 2$ for $NO_x$ sensitivity and $HCHO/NO_2 < 1$ for VOC sensitivity

(Duncan et al., 2010). These threshold values are not valid in the upper troposphere due to the vertical gradients of the trace gases. However, the $HCHO/NO_2$ ratio can also indicate the transition from a $NO_x$- to a VOC-sensitive regime when binned to NO mixing ratios, which does not require any absolute threshold values. Within the $NO_x$-sensitive regime, the $HCHO/NO_2$ strongly decreases with small increases in NO, and within the VOC-sensitive regime it is mostly unresponsive to changes in NO. Depending on where in these plots a specific data point or an average of several data points is located, it is possible to

derive the dominant chemical regime.

As explained earlier, it is not possible to determine the dominant chemical regime from ozone formation rates $P(O_3)$ in the upper troposphere, as the formation of $HNO_3$ plays a minor role at UT altitudes and therefore does not lead to a decrease in $P(O_3)$, which in theory indicates the dominance of VOC over $NO_x$ sensitivity. In fact, $P(O_3)$ does decrease for NO mixing ratios above around 0.7 ppbv, but for a different reason, as shown in Figure S10 of the Supplement. Panel (a) presents $P(O_3)$

binned to NO, which increases for low NO, reaches a plateau around 0.6–0.7 ppbv NO and decreases at higher NO. Panel (b) shows $NO_x$ loss ($L(NO_x)$) rates via OH, $HO_2$ and $CH_3O_2$, which are negligible compared to $P(O_3)$ rates as shown in panel (a). Even though $L(NO_x)$ increases with increasing NO, it is still only 6 % of the ozone production at 1 ppbv NO. The decrease

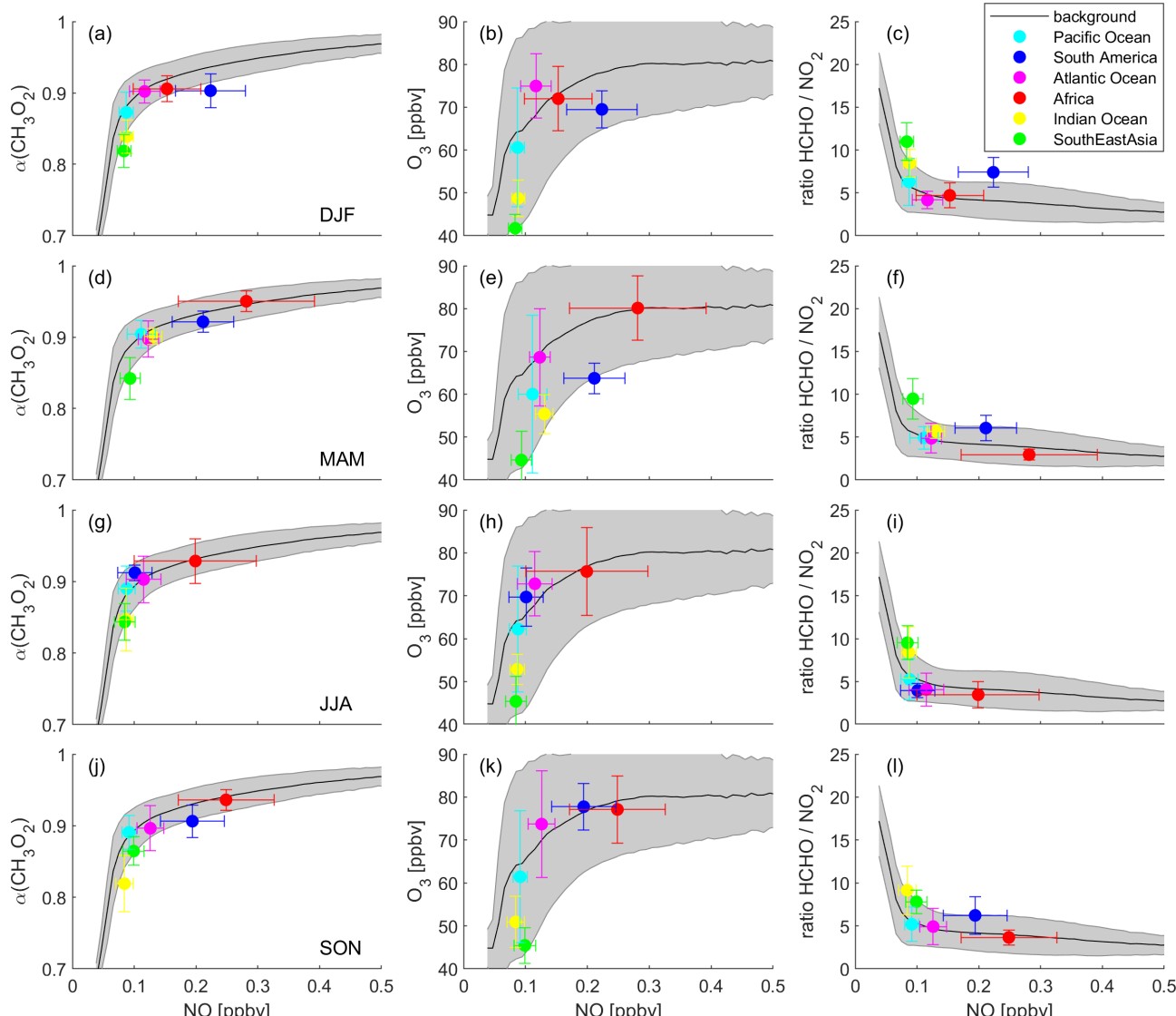

**Figure 6.** Different metrics to determine the dominant chemical regime. Left column: $\alpha(CH_3O_2)$, middle column: $O_3$ and right column: HCHO/NO$_2$ ratio, binned to NO mixing ratios for (a)-(c) DJF (December–February), (d)-(f) MAM (March–May), (g)-(i) JJA (June–August) and (j)-(l) SON (September–November). Black lines show averages of all data points and grey shades present the $1\sigma$ standard deviation. Colored data points show the averages for the indicated areas and the $1\sigma$ variability.

in P($O_3$) is therefore not associated with the formation of HNO$_3$ (as it is in the lower troposphere) but reflects the decrease of HO$_2$ with increasing NO (see panels (c)-(d)). The peak in P($O_3$), therefore, does not provide an indication for a regime change.

To investigate individual areas regarding predominant $O_3$ sensitivity, we analyze the location of the area averages along the background. Figure 6 (a) shows NO vs $\alpha(CH_3O_2)$ during DJF. The tropical UT over the Indian Ocean and South East

Asia is characterized by $NO_x$ sensitivity with NO mixing ratios between 80 and 90 pptv and an average $\alpha$ of 0.84 and 0.82, respectively. Ozone formation over South America is VOC-sensitive with an average NO mixing ratio of 222 pptv and an $\alpha$ of 0.90. The data points for the Pacific Ocean, the Atlantic Ocean and Africa are close to the transition point of the two regimes, with a tendency of the Pacific Ocean towards $NO_x$ and of the Atlantic Ocean and Africa towards VOC sensitivity. This is in line with Figure 6 (b) which presents NO vs. $O_3$ mixing ratios. The data points for South East Asia, the Indian Ocean and the Pacific Ocean are located mostly in the upsloping part of the curve, where $O_3$ strongly increases with increasing NO. The averages for the Atlantic Ocean, Africa and South America are located towards the flattening of the curve. Figure 6 (c) shows the DJF averages for NO vs. the $HCHO/NO_2$ ratio. For South East Asia, the Indian Ocean and the Pacific Ocean, NO mixing ratios are below 0.1 ppbv and $HCHO/NO_2$ ratios are high with values of 6.3, 8.5 and 10.9 ppbv ppbv$^{-1}$, respectively. For the Atlantic Ocean and Africa, the average NO mixing ratios are higher and the $HCHO/NO_2$ ratios are lower with values of 4.2 and 4.7 ppbv ppbv$^{-1}$, respectively. NO mixing ratios over South America are even higher, but the $HCHO/NO_2$ ratio is also higher with a value of 7.4 ppbv ppbv$^{-1}$. This underlines the limitation of using absolute threshold values for determining the dominant chemical regime. If a threshold for the regime transition were to be set to, e.g., 5 ppbv ppbv$^{-1}$, the South American UT would be characterized as $NO_x$-sensitive, while it clearly shows VOC sensitivity. It is therefore important to consider the metrics used in relation to ambient NO mixing ratios, and it is best to use them in combination with other metrics.

Figure 6 (d) shows $\alpha(CH_3O_2)$ binned to NO for MAM data. The UT over South East Asia is $NO_x$-sensitive with values similar to DJF. Over the Indian Ocean, both the average NO mixing ratio and $\alpha(CH_3O_2)$ increase to 130 pptv and 0.90, respectively, being located in the transition regime, together with the Pacific Ocean and the Indian Ocean. Minor changes from DJF to MAM occur over South America, which is still VOC-sensitive. A strong VOC sensitivity is calculated for the UT over Africa with average NO mixing ratios of 279 pptv and $\alpha(CH_3O_2)$ of 0.95. These findings are confirmed by $O_3$ and the $HCHO/NO_2$ ratio binned to NO in Figure 6 (e) and (f). The data for South East Asia, the Pacific Ocean, the Atlantic Ocean and South America are similar to values during DJF. Between DJF and MAM, the values over the Indian Ocean and Africa change to higher NO (131 pptv and 279 pptv) in combination with higher $O_3$ (55 ppbv and 80 ppbv) and a lower $HCHO/NO_2$ ratio (5.7 ppbv ppbv$^{-1}$ and 3.0 ppbv ppbv$^{-1}$), associated with a change from the $NO_x$-sensitive to the transition regime and a change from the transition to the VOC-sensitive regime, respectively.

Figure 6 (g), (h) and (i) show similar graphs for JJA (June–August), indicating $NO_x$ sensitivity for the UT over South East Asia and the Indian Ocean, a transition regime for the Pacific Ocean, the Atlantic Ocean and South America and VOC sensitivity for Africa. During SON (September–November), as shown in Figure 6 (j), (k) and (l), South America shifts back to VOC sensitivity. All other regimes remain unchanged between JJA and SON.

Figure S11 of the Supplement shows the mean values of the specified areas for $P(O_3)$ vs. NO. While the computational tools presented above allow for a clear distinction between the regimes depending on location and time of the year, this indicator shows no differences. According to the surface-oriented definition for chemical regimes, all data points would be located in the $NO_x$-sensitive regime.

In summary, Figure 6 illustrates three important results. First, the transition between $NO_x$ and VOC sensitivity occurs at around 0.1 ppbv NO in the upper tropical troposphere. Second, areas with increased lightning activity tend towards the

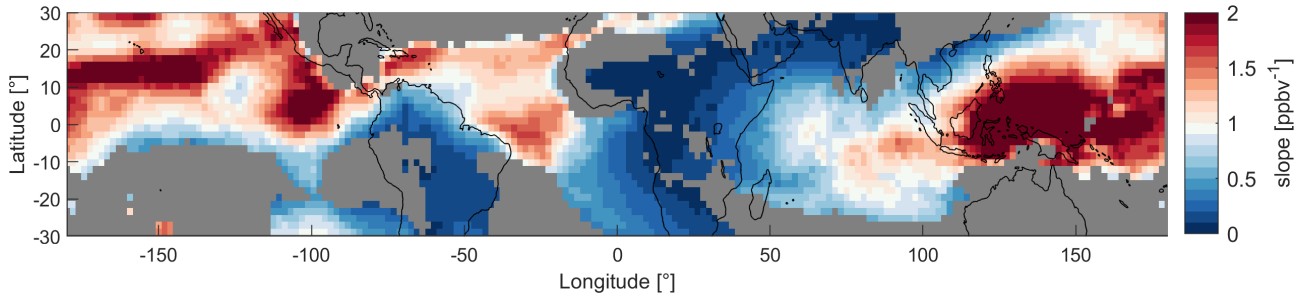

**Figure 7.** Map of the tropical UT between $30\,°\,$S and $30\,°\,$N colored by the slopes of NO vs $\alpha(CH_3O_2)$ of the data in model grid regions, exemplary for MAM. Red colors indicate $NO_x$ and blue colors VOC sensitivity. For grey areas the $R^2$ of the fit is below $30\,\%$.

dominance of VOC sensitivity. And third, the dominating regime changes with the time of the year. We will discuss these findings and their implications in the following.

Since NO and $HO_2$ mixing ratios, as well as NOPRs change throughout the year, the varying locations of deep convection
also affect the dominant chemical regime. Areas with deep convection are potentially associated with lightning activity, resulting in higher NO mixing ratios that lead to VOC sensitivity. The continental areas of South America and especially Africa experience most lightning and therefore show the most VOC-sensitive regimes (Williams and Sátori, 2004). As the cumulonimbus clouds in South East Asia are mostly formed in maritime conditions, the region experiences significantly less lightning and therefore shows $NO_x$ sensitivity all year round. Ozone formation over the oceans is either $NO_x$-sensitive or in the transition
regime as lightning strikes are significantly less frequent in maritime compared to continental areas. Figure 7 presents a geographical distribution of the tropical UT colored by the slopes of the NO vs $\alpha(CH_3O_2)$ data in each individual grid region to illustrate the dominating chemical regimes. Here, we present a map for MAM data. In Figure S12 of the Supplement, we show them for all periods. High values for the slopes and red colors (i.e. values well above 1) represent the predominantly $NO_x$-sensitive regime and low values, accompanied by blue colors represent VOC sensitivity. It is not possible to determine
an exact threshold slope for the transitioning between regimes. Generally, the more intense the color (red or blue), the more clearly a location is assigned to one of the two regimes. Lighter colors indicate a state closer to the transition regime. Grey areas indicate that the $R^2$ of the fit is $<30\,\%$ which, for example, occurs when the data are arranged in a cloud of points. This depiction is in line with the results from Figure 6. During MAM, blue colors over South America and Africa indicate a VOC-sensitive regime. Red colors over South East Asia show $NO_x$ sensitivity. Finally, over the three oceans we find lighter colors
indicating the transition regime. This view also allows for a more detailed differentiation between the areas; for example, the UT over the Atlantic Ocean tends more towards an $NO_x$-sensitive regime in the northern and towards a VOC-sensitive regime in the southern part.

While we focus on the upper troposphere in this study, $\alpha(CH_3O_2)$ remains a suitable indicator for the dominant chemical regime at all altitudes. In Figure 8 we present the slopes of NO vs $\alpha(CH_3O_2)$ by pressure altitude and longitudes, as an example
for MAM data close to the equator ($1\,°$ N). White areas at the bottom (between 800 and 1000 hPa) result from the local surface

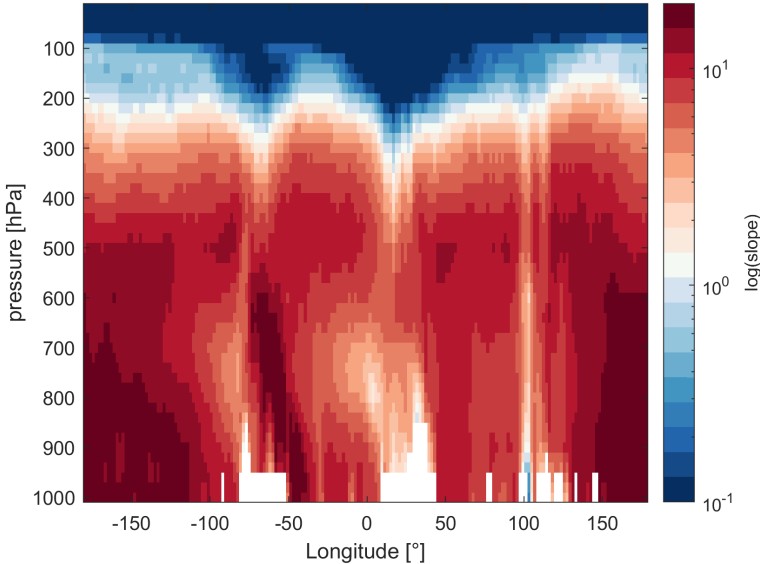

**Figure 8.** Slopes of NO vs $\alpha(\mathrm{CH_3O_2})$ by pressure altitude and longitude on a log scale during the period March–May (MAM) and close to the equator ($1\,^\circ$N). Red colors indicate $\mathrm{NO}_x$ and blue colors VOC sensitivity. White areas between 800 and 1000 hPa result from the local surface topography.

topography (mountains). For the free troposphere, we find strong $\mathrm{NO}_x$ sensitivity with a maximum slope of 38 ppbv$^{-1}$. In the upper troposphere lower stratosphere at pressure altitudes between 300 and 100 hPa, we observe the transitioning to a VOC-sensitive regime. For latitudes with strong lightning activity, including areas such as continental South America (-80 to -60 $^\circ$ longitude) and Africa (5 to 30 $^\circ$ longitude), the transition occurs in the upper troposphere, corresponding to pressure altitudes of 250–300 hPa. For latitudes with low lightning activity, for example, between 130 and 160 $^\circ$ longitude (South East Asia), the regime change only occurs at the transition to the lower stratosphere – at a pressure altitude of around 150 hPa – which is characterized by strong $\mathrm{NO}_x$ saturation. In Figure S13 of the Supplement we additionally show the dominant chemical regime, indicated by NO vs $\alpha(\mathrm{CH_3O_2})$, on a global scale near the surface as the annual average. As we would expect, $\alpha(\mathrm{CH_3O_2})$ indicates VOC sensitivity at the surface for all urbanized and industrialized regions characterized by high $\mathrm{NO}_x$ emissions and $\mathrm{NO}_x$ sensitivity for remote regions. Shipping routes, which are closer to the transition regime, can be distinguished from the pronounced maritime $\mathrm{NO}_x$ sensitivity.

### 3.3.2 Sensitivity study: lightning $\mathrm{NO}_x$

Three additional model runs were performed in order to investigate the impact of lightning $\mathrm{NO}_x$. First, lightning $\mathrm{NO}_x$ was completely omitted. In second and third runs, $\mathrm{NO}_x$ from lightning was halved and doubled, respectively, compared to the baseline scenario. The emissions of global lightning $\mathrm{NO}_x$ in the baseline scenario amount to 6.2 Tg/year (estimated from the climatological data) in agreement with the work of Miyazaki et al. (2014).

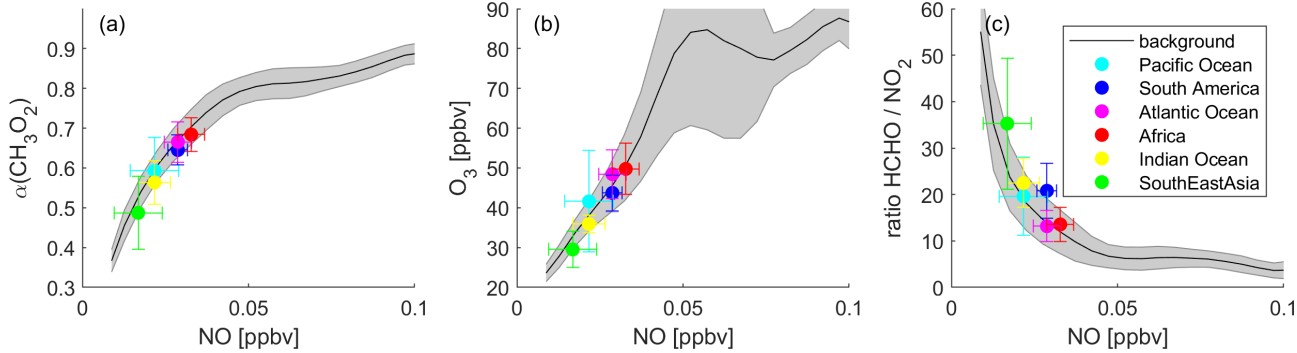

**Figure 9.** Determination of the dominant chemical regime in the tropical UT in the "no lightning" scenario via (a) $\alpha(CH_3O_2)$, (b) $O_3$ and (c) the HCHO/NO$_2$ ratio, binned to NO mixing ratios. Black lines show averages of all data points and grey shades present the $1\sigma$ standard deviation. Colored data points show the averages for the indicated areas with the $1\sigma$ variability.

Figure 9 shows the three previously discussed metrics (a) $\alpha(CH_3O_2)$, (b) $O_3$ and (c) the HCHO/NO$_2$ ratio binned to NO mixing ratios for the modeling scenario excluding lightning NO$_x$. As there are no significant differences between the periods, we show all-year data here. Figure S14 of the Supplement shows the subdivision into the four periods (DJF, MAM, JJA and SON), and in Figure S15, we present a comparison between the baseline scenario as a yearly average and the scenario excluding lightning. The black lines representing the average of all data points show a similar course compared to the baseline scenario including lightning NO$_x$, but the distinction between the regimes is less pronounced. Figure 9 (a) presents NO vs $\alpha(CH_3O_2)$. At low NO mixing ratios, $\alpha(CH_3O_2)$ increases with NO, indicating NO$_x$ sensitivity and for higher NO mixing ratios, $\alpha(CH_3O_2)$ is only marginally affected by changes in NO, indicating VOC sensitivity. The tropical UT over all selected areas is clearly located within the NO$_x$-sensitive chemical regime. The average NO mixing ratios range from 17 pptv over South East Asia to 33 pptv over Africa. Compared to the baseline scenario, excluding lightning NO$_x$ leads to a decrease of ambient NO levels by up to one order of magnitude. The average $\alpha(CH_3O_2)$ ranges from 0.49 to 0.68 over South East Asia and Africa, respectively. The abundance of HO$_2$ in comparison to NO is therefore high, and a significant amount of CH$_3$O$_2$ undergoes reaction with HO$_2$, next to NO. Figure 9 (b) shows O$_3$ mixing ratios as a function of NO. O$_3$ mixing ratios first increase as a function of NO, reach a peak at around 0.05 ppbv NO and 85 ppbv O$_3$ and subsequently change little at higher NO levels. Note that the number of data points decreases rapidly for high NO mixing ratios. Only around 5.5 % of the data points represent NO values of $> 0.05$ ppbv. We show the associated frequency distribution in Figure S16 of the Supplement. As expected, the data points of all selected areas are located at low NO and O$_3$ levels, at average O$_3$ mixing ratios ranging from 30 ppbv over South East Asia to 50 ppbv over Africa. Figure 9 (c) shows the HCHO/NO$_2$ ratio binned to NO mixing ratios. The course of the average values (black line) is again similar to the one for the baseline scenario, but the values for the HCHO/NO$_2$ ratio are higher. The highest average value occurs over South East Asia with 35.2 ppbv ppbv$^{-1}$ and the lowest over the Atlantic Ocean with 13.2 ppbv ppbv$^{-1}$. All data points are therefore clearly located within the NO$_x$-sensitive regime, which is in line with the findings from the correlation between NO and $\alpha(CH_3O_2)$ from Figure 9 (a).

In Figure S17 and S18, we present the three metrics $\alpha(CH_3O_2)$, $O_3$ and the $HCHO/NO_2$ ratio binned to NO mixing ratios for

all periods and locations for halved and doubled lightning $NO_x$, respectively. The transition region between the regimes occurs around 0.1 ppbv and is therefore not meaningfully different from that in the baseline scenario, but the distinction between the regimes is more conspicuous with increasing lightning $NO_x$ emissions.

Figure 10 shows the average $\alpha(CH_3O_2)$ vs. NO for each considered location and for all modeled lightning $NO_x$ scenarios. As expected, in each location the data points shift to higher values for both NO and $\alpha(CH_3O_2)$ with increasing influence of

lightning $NO_x$. When excluding lightning, the dominant regime changes to $NO_x$-sensitive in all locations. Removing lightning also shows that lightning is by far the dominant source of $NO_x$ in the upper tropical troposphere. In maritime regions where lightning is relatively infrequent, $NO_x$ depends more strongly on advection from continental regions, formation from $HNO_3$ and aircraft emissions. A model run excluding $NO_x$ emissions from aircraft does not lead to significant differences compared to the baseline scenario, which we present by the black crosses in Figure 10. This may contradict recent findings by Wang et al.

(2022a), who presented increases in upper tropospheric ozone in response to increasing aircraft emissions. In our model, $NO_x$ in the tropical upper troposphere is dominated by lightning emissions, whereas $NO_x$ concentrations from aviation in this part of the troposphere are small, with most aircraft emissions occurring north of $30\,^\circ$N latitude. Excluding lightning shows that $NO_x$ mixing ratios also decrease significantly in maritime environments, including South East Asia where $NO_x$ mixing ratios drop from 90 pptv to 17 pptv on average. This illustrates that in maritime locations in the tropics, i.e., apart from South America

and Central Africa, $NO_x$ mixing ratios are largely dependent on transported lightning $NO_x$. For halved lightning $NO_x$, $NO_x$ sensitivity also prevails in most locations. Only Africa and South America show a transition regime for the periods of the year with maximum lightning. For doubled lightning $NO_x$, the qualitative regime observations are similar to the baseline scenario. The UT over Central Africa and South America is mostly VOC-sensitive, over South East Asia and the Indian Ocean it is $NO_x$-sensitive and this layer is in the transition regime over the Pacific and Atlantic Ocean. Therefore, regions with frequent

lightning are VOC-sensitive in the baseline scenario while the doubling of lightning $NO_x$ does not have a large impact in regions where lightning is generally infrequent. In accordance with our prior analysis, $O_3$ does not increase significantly from the doubling of lightning $NO_x$. In the VOC-sensitive regime, the black curve representing the average of all data points of NO vs $O_3$ levels off at around 90 ppbv compared to 80 ppbv for the baseline scenario. This aids our understanding of $NO_x$ and VOC sensitivity in the upper troposphere, as all available $HO_2$ and $CH_3O_2$ radicals react with NO within the VOC-sensitive

regime and changes in $NO_x$ therefore do not affect changes in $O_3$.

The sensitivity study of lightning $NO_x$ emphasizes two major aspects. First, lightning is the predominant source of $NO_x$ in the upper tropical troposphere as the mixing ratios drop to near zero when excluding it, and a model run excluding aircraft $NO_x$ does not show significant differences compared to the baseline scenario. Second, lightning and its distribution in the tropics, which is affected by the partitioning of continental and maritime areas and the varying locations of deep convection throughout

the year, are the most important determinants of the dominant chemical regime in the UT. Our results additionally indicate that any future changes in lightning will only significantly affect $O_3$ levels in the upper troposphere if lightning substantially increases in locations where it is currently sparse or if lightning decreases in areas where it is presently frequent. Our results fit well with previous literature on the role of lightning $NO_x$ in $O_3$ production in the upper troposphere. Grewe (2007) reported

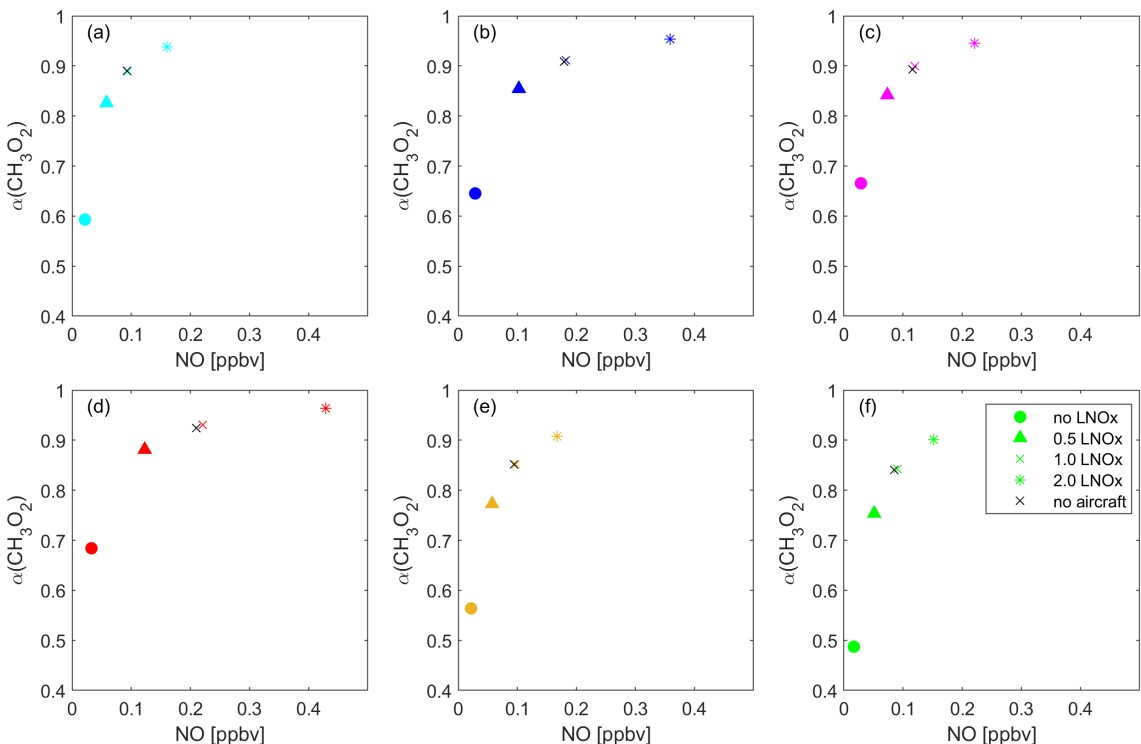

**Figure 10.** Overview of the impact of lightning $NO_x$ on the dominant $O_3$ sensitivity. The data points present the average NO vs $\alpha(CH_3O_2)$ of all data points located in each of the six regions (a) Pacific Ocean (cyan), (b) South America (blue), (c) Atlantic Ocean (pink), (d) Africa (red), (e) Indian Ocean (yellow), (f) South East Asia (green) for the baseline scenario (crosses), excluding lightning (circles), halved lightning (triangles) and doubled lightning (asterisks).

that $NO_x$ from lightning is the dominant source of $O_3$ in the upper tropical troposphere based on simulations with the global climate–chemistry model E39/C. Similar results have been presented by Murray (2016), Schumann and Huntrieser (2007) and Sauvage et al. (2007). While it has been shown previously that ozone production in the UTT is highly dependent on lightning $NO_x$, this study is the first to extensively study the impact of lightning on the dominant $O_3$ regime, applying a new indicator $\alpha(CH_3O_2)$, which is valid throughout the entire troposphere.

## 4 Conclusions

We have investigated the dominance of $NO_x$ and VOC sensitivity in the upper tropical troposphere (200 hPa) between $30\,°$ S and $30\,°$ N latitude. The analyzed trace gas mixing ratios and meteorological parameters are calculated with the EMAC atmospheric chemistry - general circulation model for a $1.875\,°$ x $1.875\,°$ horizontal resolution and the years 2000–2019. One model run considers a baseline scenario and four additional ones were run with halved, doubled and excluded lightning $NO_x$ emissions, as well as excluded $NO_x$ aircraft emissions. We find that the mixing ratios of the considered trace gases have not changed

significantly in the upper troposphere over the past two decades and we therefore evaluate the average of the data which benefits from a higher statistical significance. The distribution of the analyzed trace gases varies with the time of the year and the changing areas of deep convection, confined within the ITCZ. During DJF, maximum convection occurs over South America, Central to South Africa and North Australia and during JJA, over Central America, North Africa and continental Asia. As a consequence, $NO_x$ mixing ratios and net ozone production rates peak over South America, South Africa and North

Australia during DJF, over South America and Central Africa during MAM and SON, and over Central America and North Africa during JJA, as deep convection brings increased thunderstorm and lightning activity, particularly over continental areas. The distribution of $HO_2$ mostly differs from $NO_x$ due to enhanced mixing ratios over South East Asia, where $NO_x$ is low year around.

   We analyzed several commonly applied metrics for their potential to determine the dominant chemical regime in the upper

troposphere, including ozone production rates $P(O_3)$, the fraction of methyl peroxyradicals forming formaldehyde $\alpha(CH_3O_2)$, and the ratio of HCHO to $NO_2$. We show that $\alpha(CH_3O_2)$ and the $HCHO/NO_2$ are good indicators for the chemical regime in the upper troposphere, while $P(O_3)$ is unsuitable. At the surface, $NO_x$ sensitivity is generally defined by increasing $P(O_3)$ with NO, and VOC sensitivity by decreasing $P(O_3)$ with NO. In the upper troposphere, this indicator is no longer valid as the reaction of $NO_2$ with OH does not play a significant role. Instead, under conditions of $NO_x$ sensitivity $CH_3O_2$ undergoes

reaction with both $HO_2$ and NO, and increasing NO leads to an enhancement of $O_3$. For VOC-sensitive conditions, $CH_3O_2$ predominantly reacts with NO and as the latter is present in excess it does not influence $O_3$ mixing ratios. In this case, ozone formation changes are governed by those in VOC, controlling the availability of peroxy radicals. The transition point can be read from the course of $\alpha(CH_3O_2)$ and the $HCHO/NO_2$ ratio as a function of NO abundance. This definition of chemical regimes in terms of $NO_x$ and VOC sensitivity is valid throughout the entire troposphere. When assessing $O_3$ sensitivity in

the upper troposphere based on trace gas measurements, $\alpha(CH_3O_2)$ is to be preferred over the $HCHO/NO_2$ ratio as it can be more easily determined from in-situ data. While NO and $HO_x$ measurements are commonly performed on research aircraft, for example, $NO_2$ measurements tend to suffer from the unselective detection or artifacts from reservoir species, which makes accurate quantification challenging (Reed et al., 2016; Jordan et al., 2020; Andersen et al., 2021; Nussbaumer et al., 2021b).

   In the ITCZ over continental areas, ozone chemistry is mostly VOC-sensitive. The UT over South America and Africa

is therefore VOC-sensitive apart from JJA and DJF, respectively, where chemistry moves towards the transition area. Over maritime areas, including South East Asia, ozone chemistry is mostly $NO_x$-sensitive or in the transition regime depending on the time of the year. The metrics which are found to be good indicators for the UT, $\alpha(CH_3O_2)$, $O_3$ mixing ratios and the $HCHO/NO_2$ ratio as a function of NO, show that the transition between a $NO_x$- and a VOC-sensitive regime occurs around 0.1 ppbv NO. When decreasing or excluding lightning $NO_x$, the considered areas are mostly dominated by a $NO_x$-sensitive

regime. We can therefore conclude that lightning is the major driver of the dominating ozone sensitivity in the upper tropical troposphere. While it is still not fully understood how lightning activity will evolve in the future, it remains important to monitor and understand ozone production in the upper tropical troposphere, a process which has a major impact on the radiative energy budget, and in turn on global warming.

*Data availability.* Model data will be uploaded to a public data repository upon acceptance of the manuscript.

*Author contributions.* CMN, HF and AP conceived the study. CMN analyzed the data and wrote the manuscript. AP provided the modeling data. All authors contributed to designing the study and proofreading the manuscript.

*Competing interests.* At least one of the (co-)authors is a member of the editorial board of Atmospheric Chemistry and Physics.

*Acknowledgements.* This work was supported by the Max Planck Graduate Center (MPGC) with the Johannes Gutenberg-Universität Mainz.

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
