# Peer review of "What controls ozone sensitivity in the upper tropical troposphere?"

_EGUsphere, 2023_

## Author Comment (AC1)

**Referee 1**

This paper provides a detailed analysis of ozone chemical regimes across the tropics, with a focus on the role of short-lived VOCs.  The paper is well written and the analysis is thorough and it provides new insight, but my main concern is that it is missing the big-picture by focusing too much on the details of VOCs.  Methane plays a huge role in global (and tropical) ozone production, especially in remote regions dominated by lightning NOx.  Methane is increasing rapidly and it is expected to increase over the next few decades, and therefore it will continue to drive ozone production across the tropics, even if NOx emissions were to remain constant.  Methane cannot be ignored and this paper needs to consider the role of methane as the background driver of tropical ozone production.  I provide more details below, and I touch on a few other items that need to be addressed.

We thank the referee for the time to review our manuscript and for the valuable feedback. We fully agree with the referee that methane plays an important role in $O_3$ production and we regret that this does not seem to be clear enough in the manuscript. Methane forms $O_3$ within the catalytic $HO_x$ cycle, initiated by the reaction with OH radicals. Inevitably, $CH_4$ forms methyl peroxy radicals $CH_3O_2$ as an intermediate on its way to $O_3$. And this is precisely the major advantage of the metric $\alpha(CH_3O_2)$ that we are employing to investigate the sensitivity of $O_3$ towards its precursors. It does not require detailed knowledge about VOCs, but we consider the role of VOCs indirectly through their oxidized intermediates. In fact, we show that $CH_3O_2$ represent the predominant portion of $RO_2$ radicals in the upper troposphere which acknowledges and underlines that $CH_4$ is one, if not the most crucial VOC involved in $O_3$ production in the UTT. When calculating $\alpha(CH_3O_2)$, $CH_3O_2$ mixing ratios truncate out of the fraction, and the influence on $O_3$ sensitivity simplifies to the availability of NO and $HO_2$ radicals. More important than the abundance of methane for $O_3$ sensitivity is the abundance of $HO_2$ radicals, whose availability depends partly on $CH_4$, but also on OH, CO, other VOCs and $O_3$.

We have added text in the manuscript in various locations in order to clarify this very important point.

Lines 15 ff.: We show that effectively only the knowledge of the availability of NO and $HO_2$ is required to adequately represent $O_3$ precursors and its sensitivity towards them.

Lines 166 ff.: The formation of HCHO from $CH_3O_2$ through reaction with NO leads to $O_3$ formation, as $NO_2$ is formed simultaneously, which can subsequently react to $O_3$ via photolysis. In contrast, the reaction of $CH_3O_2$ with $HO_2$ represents a termination reaction of the $HO_x$ cycle and therefore decelerates $P(O_3)$. $CH_3O_2$ here is a proxy for VOCs which form HCHO and $O_3$ through $RO_2$. A principal precursor to $CH_3O_2$ is $CH_4$, which is oxidized by OH radicals and reacts with $O_2$ in the first two steps of the catalytic $HO_x$ cycle. The atmospheric $CH_4$ concentration is increasing rapidly, with the tropics implicated, suggestive of feedbacks that may accelerate $CH_4$ growth in future contributing to $O_3$ formation in the upper troposphere (Griffiths et al., 2021; Nisbet et al., 2023) Other precursors to $CH_3O_2$ can be acetone, methyl hydroxy peroxide or acetaldehyde through photolysis or reaction with OH radicals (Nussbaumer et al., 2021a). The advantage of considering $CH_3O_2$ is that it represents an entire group of

VOCs that can form $O_3$, including CH4, instead of handling multiple trace gases and risking incompleteness.

Lines 188 ff.: This method underlines that $O_3$ sensitivity is dependent on the availability of NO and $HO_2$ radicals. Individual VOCs do not need to be considered when investigating $O_3$ sensitivity, as $HO_2$ effectively represents their chemical impact.

Methane is a major ozone precursor especially in the remote atmosphere and it will play a major role in future ozone increases across the tropics (Young et al., 2013), yet methane is not even discussed in this paper. Methane cannot be ignored and needs to be addressed. Please consider the following:

Zhang et al. (2016) ran the CAM-Chem model for 1980-2010 and estimated an increase of the global tropospheric ozone burden of 28.12 Tg (8.9%), due to the increase of anthropogenic emissions and the partial shift of the emissions from mid-latitudes towards the equator. The increase of methane (15% over 30 years) accounted for one quarter of the ozone burden increase. The increase of methane has continued to the present, with rapid increases over the past 5-10 years (https://gml.noaa.gov/ccgg/trends_ch4/). Under a future scenario of high anthropogenic emissions and continuously increasing methane concentrations (Griffiths et al., 2021), the global ozone burden is expected to increase for the remainder of the 21st century (see the ssp370 scenario in Figure 6.4 of Szopa et al., 2021), with increases of approximately 10% from 2014 to 2050.

If we understand correctly, the values presented in Zhang et al. (2016) relate to the entire tropospheric column (not only to the upper troposphere) and we therefore believe these results are not inconsistent with our results. From Figure 3d from Zhang et al. (2016) we understand that $CH_4$ did not relevantly affect the $O_3$ change between 1980 and 2010 in the upper troposphere (200hPa ~ 11-13km), where lightning is effective. These findings are in line with our observations of a relatively small increase in $CH_4$ in the upper tropical troposphere. Figure 6.4 of Szopa et al. (2021) seems to relate to the overall tropospheric $O_3$ burden as well. Regarding the change of methane over time, our model simulations are not addressing the increase over the last two decades (which may be underestimated). We would like to emphasize that this study does not aim to investigate the change over time in $O_3$ precursors. We do find an increase in average $O_3$ from ~65ppbv to ~70ppbv which corresponds to an increase by ~8% and is in line with the value stated by Zhang et al. (2016), however this change is not significant considering the variability of the order of 30 ppbv (~40-50%). This large variability likely arises from the broad geographic area that we are investigating. Additionally, despite the rise in methane level over the past decades, the $HO_x$ budget in the upper tropical troposphere has remained relatively constant, also indicated by the fact that the methane lifetime has not changed relevantly over time (Lelieveld et al. (2016), Naik et al. (2013)). From the perspective of VOC, a change in $O_3$ sensitivity is induced by changes in $HO_2$. Therefore, we expect the changes in methane over the past 20 years to have a negligible impact on the dominating $O_3$ regime.

In terms of ozone production from lightning, methane is a key precursor, first explored above the tropical South Atlantic by Moxim and Levy, 2000. They found that the upper tropospheric ozone maximum above this region was dominated by ozone production from lightning NOx in conjunction with CO/CH4 chemistry. They

suggested an ozone maximum would have existed in this region in preindustrial times. Similarly, Cooper et al. (2006) calculated that 69–84% (11–13 ppbv) of the observed upper-tropospheric ozone enhancement above North America during summer 2004 was due to in situ ozone production from lightning NOx and background mixing ratios of CO and CH4. In situ observations from the NASA DC8 showed very low values of reactive hydrocarbons in the UT, indicating a limited impact from fresh surface emissions. Basically, given plenty of lightning NOx, background methane concenrations and sunny conditions in the tropics, short-lived VOCs aren't required to produce large amounts of ozone.

We agree with the referee and we have added these references to support our findings.

Lines 40 f.: Previous studies have shown that methane (CH$_4$) is one of the most important VOC precursors to O$_3$ in the upper troposphere (Moxim and Levy, 2000; Cooper et al., 2006).

The current authors conducted sensitivity tests to understand the impact of lightning NOx, and it would be very informative if they can estimate future changes in ozone with increasing methane. Given that methane has increased almost continuously since the 1980s, and given the high likelihood that it will continue to increase, testing the sensitivity of the model to increased methane should be a priority.

We disagree with the referee regarding the need to perform a sensitivity study for methane. As discussed above, more important than the availability of methane for investigating O$_3$ sensitivity is the abundance of HO$_2$ radicals. As significant changes in HO$_2$ are unlikely in the near future, we believe a sensitivity study regarding CH$_4$ is outside the scope of this manuscript.

Line 40

"…model run excluding aircraft NOx does not show significant differences compared to the baseline scenario". This result contradicts the recent findings of Wang et al. (2022) who concluded that increasing aircraft emissions are playing a major role in increasing the global tropospheric ozone burden. Please address this discrepancy.

The applicability of the finding of Wang et al. (2022) to the upper tropical troposphere is ambiguous. Actually, this is unlikely because aircraft emissions of NO$_x$ are most important north of 30°N latitude. The share of NO$_x$ emissions from aircraft should be small in comparison to NO$_x$ originating from lightning in the upper tropical troposphere. According to Schumann and Huntrieser (2007), the fraction of aircraft NO$_x$ is close to 7% between 35°S and 35°N (assuming only sources in the UTT are aircraft and lightning) between 1995-2000. Even if these values doubled up to 2019 (Wang et al. 2022 estimated 76% between 1995 and 2017) and lightning remained constant, the fraction of NO$_x$ from aircraft would still be low. Based on our findings, around 5% of NO$_x$ originates from aircraft emissions over Africa, South East Asia and the Atlantic Ocean, and even less in the remaining areas, looking at the period from 2000-2019. When O$_3$ chemistry is NO$_x$ sensitive, small changes in NO$_x$ can impact O$_3$ levels, but when it is VOC sensitive due to strong lightning, e.g. over Africa or South America, we do not expect aircraft to have a significant impact on O$_3$. We added text to highlight this discrepancy with Wang et al. (2022).

Lines 444 f.: This may contradict recent findings by Wang et al. (2022), who presented increases in upper tropospheric ozone in response to increasing aircraft emissions. In our model, $NO_x$ in the tropical upper troposphere is dominated by lightning emissions, whereas $NO_x$ concentrations from aviation in this part of the troposphere are small, with most aircraft emissions occurring north of 30°N latitude.

Line 26

"In the upper troposphere (UT), ozone is the second most important greenhouse gas after water vapor and changes in ozone exert (and will continue to exert) a particularly large impact on the earth's radiative forcing – especially in the tropopause region and the tropical UT (Lacis et al., 1990; Mohnen et al., 1993; Wuebbles, 1995; Lelieveld and van Dorland, 1995; van Dorland et al., 1997; Staehelin et al., 2001; Iglesias-Suarez et al., 2018)."

On the global scale, ozone is the third most important greenhouse gas after CO2 and methane (not including water vapor) and so I'm surprised by this statement that ozone has a greater impact than CO2 and methane on the radiative balance of the UT. I looked at the list of references and none of them seem to make a strong case for this claim. Perhaps this was true decades ago when CO2 and methane concentrations were much lower, but they have increased greatly since the study by Lacis et al. 1990.

The NOAA Annual Greenhouse Gas Index (https://gml.noaa.gov/aggi/aggi.html) has increased by 49% since 1990, driven by increases in methane and CO2 and corresponding to a radiative forcing increase from 2.3 W m-2 to 3.4 W m-2. The index does not include ozone, but the increase in radiative forcing due to ozone from 1990 to 2014 is only about 0.1 W m-2. Figure 1 in the supplement to Skeie et al. (2020) has the most recent update on the height/latitude distribution of ozone's radiative forcing. Do you have any similar plots to compare it to, which would indicate if ozone has a stronger radiative effect in the tropical UT than CO2 or methane?

**Thanks for bringing this to our attention. Indeed, our statement was incorrect. We have adjusted the text accordingly and added Skeie et al. (2020) as a reference.**

Lines 27 ff.: Ozone is the third most important anthropogenic greenhouse gas after carbon dioxide ($CO_2$) and methane ($CH_4$), with a particularly strong impact in the upper troposphere where concentrations of the natural greenhouse gas water vapor are small compared to the surface. Changes in ozone exert (…).

and

Line 2: Ozone is an important contributor to the radiative energy budget of the upper troposphere (UT).

References:

Cooper, O. R., A. Stohl, M. Trainer, A. Thompson, J. C. Witte, S. J. Oltmans, G. Morris, K. E. Pickering, J. H. Crawford, G. Chen, R. C. Cohen, T. H. Bertram, P. Wooldridge, A. Perring, W. H. Brune, J. Merrill, J. L. Moody, D. Tarasick, P. Nédélec, G. Forbes, M. J. Newchurch, F. J. Schmidlin, B. J. Johnson, S. Turquety, S. L.

Baughcum, X. Ren, F. C. Fehsenfeld, J. F. Meagher, N. Spichtinger, C. C. Brown, S. A. McKeen, I. S. McDermid and T. Leblanc (2006), Large upper tropospheric ozone enhancements above mid-latitude North America during summer: In situ evidence from the IONS and MOZAIC ozone monitoring network, J. Geophys. Res., 111, D24S05, doi:10.1029/2006JD007306.

Griffiths, P. T., Murray, L. T., Zeng, G., Shin, Y. M., Abraham, N. L., Archibald, A. T., Deushi, M., Emmons, L. K., Galbally, I. E., Hassler, B., Horowitz, L. W., Keeble, J., Liu, J., Moeini, O., Naik, V., O'Connor, F. M., Oshima, N., Tarasick, D., Tilmes, S., Turnock, S. T., Wild, O., Young, P. J., and Zanis, P.: Tropospheric ozone in CMIP6 simulations, Atmos. Chem. Phys., 21, 4187–4218, https://doi.org/10.5194/acp-21-4187-2021, 2021.

Moxim, W. J., and H. Levy II (2000), A model analysis of the tropical South Atlantic Ocean tropospheric ozone maximum: The interaction of transport and chemistry, J. Geophys. Res., 105, 17,393– 17,415

Skeie, R.B., Myhre, G., Hodnebrog, Ø., Cameron-Smith, P.J., Deushi, M., Hegglin, M.I., Horowitz, L.W., Kramer, R.J., Michou, M., Mills, M.J. and Olivié, D.J., 2020. Historical total ozone radiative forcing derived from CMIP6 simulations. Npj Climate and Atmospheric Science, 3(1), p.32.

Szopa, S., V. Naik, B. Adhikary, P. Artaxo, T. Berntsen, W.D. Collins, S. Fuzzi, L. Gallardo, A. Kiendler-Scharr, Z. Klimont, H. Liao, N. Unger, and P. Zanis, 2021: Short-Lived Climate Forcers. In Climate Change 2021: The Physical Science Basis. Contribution of Working Group I to the Sixth Assessment Report of the Intergovernmental Panel on Climate Change [Masson-Delmotte, V., P. Zhai, A. Pirani, S.L. Connors, C. Péan, S. Berger, N. Caud, Y. Chen, L. Goldfarb, M.I. Gomis, M. Huang, K. Leitzell, E. Lonnoy, J.B.R. Matthews, T.K. Maycock, T. Waterfield, O. Yelekçi, R. Yu, and B. Zhou (eds.)]. Cambridge University Press, Cambridge, United Kingdom and New York, NY, USA, pp. 817–922, doi:10.1017/9781009157896.008

Wang, H., et al. (2022), Global tropospheric ozone trends, attributions, and radiative impacts in 1995–2017: an integrated analysis using aircraft (IAGOS) observations, ozonesonde, and multi-decadal chemical model simulations, Atmos. Chem. Phys., 22, 13753–13782, https://doi.org/10.5194/acp-22-13753-2022

Young, P. J., Archibald, A. T., Bowman, K. W., Lamarque, J.-F., Naik, V., Stevenson, D. S., Tilmes, S., Voulgarakis, A., Wild, O., Bergmann, D., Cameron-Smith, P., Cionni, I., Collins,W. J., Dalsøren, S. B., Doherty, R. M., Eyring, V., Faluvegi, G., Horowitz, L. W., Josse, B., Lee, Y. H., MacKenzie, I. A., Nagashima, T., Plummer, D. A., Righi, M., Rumbold, S. T., Skeie, R. B., Shindell, D. T., Strode, S. A., Sudo, K., Szopa, S., and Zeng, G.: Preindustrial to end 21st century projections of tropospheric ozone from the Atmospheric Chemistry and Climate Model Intercomparison Project (ACCMIP), Atmos. Chem. Phys., 13, 2063–2090, https://doi.org/10.5194/acp-13-2063-2013, 2013.

Zhang, Y., et al. (2016), Tropospheric ozone change from 1980 to 2010 dominated by equatorward redistribution of emissions, Nature Geoscience, 9(12), p.875, doi: 10.1038/NGEO2827.

---

## Author Comment (AC2)

**Referee 2**

This paper uses a 20-year simulation of the EMAC model to demonstrate the reliability of different metrics for defining $NO_x$ and VOC-sensitive regimes for ozone production in the tropical upper troposphere. They find that these chemical regimes are well defined using the formaldehyde to $NO_2$ ratio ($HCHO/NO_2$) and the production of HCHO from methyl peroxyradicals ($a(CH_3O_2)$), parametrized as a function of NO mixing ratios. The sensitivity of the chemical regimes to changes in lighting $NO_x$ production is presented as a case study. Overall, the paper is well written and within the scope of ACP. Understanding the drivers of ozone production in the upper troposphere is important for projecting the role of ozone in climate change.

We thank the referee for taking the time to review our manuscript and the positive feedback.

However, discussion should be added explaining how the results of the present study fit in with earlier publications. Additionally, given the role of lightning $NO_x$ and aircraft emissions in the paper, how these emissions are implemented in EMAC should be described in Section 2.3. Please also explain how the 200 hPa pressure level was chosen for this analysis and if it might have an impact on the results, such as the relative importance of lightning NOx and aircraft emissions. In general, the paper spends a lot of time listing details that are summarized in the figures at the expense of communicating the meaning of the results. Examples are given under the specific comments where more concise language would be beneficial.

We have added discussion in view of earlier publications.

Lines 467 ff.: Our results fit well with previous literature on the role of lightning $NO_x$ in $O_3$ production in the upper troposphere. Grewe (2007) reported that $NO_x$ from lightning is the dominant source of $O_3$ in the upper tropical troposphere based on simulations with the global climate–chemistry model E39/C. Similar results have been presented by Murray (2016), Schumann and Huntrieser (2007) and Sauvage et al. (2007). While it has been shown previously that ozone production in the UTT is highly dependent on lightning $NO_x$, this study is the first to extensively study the impact of lightning on the dominant $O_3$ regime, applying a new indicator $\alpha(CH_3O_2)$, which is valid throughout the entire troposphere.

We have added some information on the implementation of emissions of $NO_x$ from lightning and aircraft in Section 2.3.

Lines 213 ff.: In this work the emissions of $NO_x$ from lightning are following the work of Tost et al. (2007). A constant $NO_x$ emission of ~20,6 kg N/flash is used for cloud-to-ground flashes. The ratio of $NO_x$ production by intra-cloud flashes is lower by a factor of 0.1. The lightning frequency is estimated with the parameterization of Grewe et al. (2001). Here the updraft velocity (used as proxy for convective strength) is associated with cloud electrification and therefore proportional to frequency of the lightning flashes. On the other side, aircraft emissions are prescribed from the CAMS dataset (CAMS-GLOB-AIR v1.1, Garnier et al, 2019).

The 200hPa pressure level was chosen to represent upper tropical tropospheric influence. At this altitude, we capture lightning events, but is not strongly influenced

by transport from the stratosphere. This is also highlighted in Figure 8 of our manuscript. At lower altitudes in the free troposphere, $O_3$ chemistry is strongly $NO_x$ sensitive due to the absence of NOx sources. In contrast, in the stratosphere $O_3$ chemistry is strongly VOC sensitive. The UTT at ~200hPa marks the transition area between $NO_x$ and VOC sensitivity, which is strongly impacted by lightning. We have added this information to the text.

Lines 209 ff.: This pressure level was chosen to represent upper tropospheric influence in the tropics in order to capture lightning events and avoid strong influence of transport from the stratosphere. The choice is also underlined by the results presented in Figure 7 where 200 hPa marks the area of transition from $NO_x$ sensitivity in the free troposphere to VOC sensitivity in the stratosphere, impacted by lightning.

**Specific Comments:**

- In the introduction, please write out chemical reactions that are described and used again in the paper or are important for understanding the regime metrics.

  We have added important chemical reactions. For the $HO_x$ cycle, we kindly ask the reader to refer to the stated literature as it has been published and discussed in detail many times before.

  Lines 45 f.: Deviations from the $HO_x$ cycle, including self-reactions of peroxy radicals, (Reaction (R2)), and the reaction of OH with $NO_2$ forming $HNO_3$ (Reaction (R3)), can terminate the formation of ozone.

  $$HO_2 + HO_2 \rightarrow H_2O_2 + O_2 \quad RO_2 + HO_2 \rightarrow ROOH + O_2 \quad RO_2 + RO_2 \rightarrow ROOR + O_2 \qquad (R2)$$

  $$NO_2 + OH \rightarrow HNO_3 \qquad (R3)$$

- Line 44: In terms of surface $O_3$, knowing if a region is $NO_x$ or VOC sensitive is important for determining regulatory policies for controlling air quality. Please state why knowing the regime is useful for the upper troposphere and connect these concepts to the first two paragraphs of the introduction. This is mentioned later, but motivation should be given upfront.

  We have added some text to underline the importance of investigating dominating $O_3$ sensitivity at high altitudes.

  Lines 58 ff.: Due to its wide-ranging atmospheric effects, not only on air quality and human health at the surface but also on atmospheric oxidation processes (the self-cleaning mechanism) and on climate change, with the upper troposphere being especially sensitive, it is highly relevant to understand and monitor $O_3$ levels and the main drivers of its sensitivity also at these altitudes.

- Line 47, "…most of the indicators for either regime are no longer valid.": Please provide a brief explanation why. This sentence might make more sense if it was moved to the description of indicators that are not suitable for the upper troposphere.

Thanks for pointing this out. We have added some text for clarification.

Lines 54 ff.: These metrics are well studied and reported at surface levels. However, most of the indicators for either regime are no longer valid when it comes to the upper troposphere. This is due, for example, to the changing ratios of the investigated trace gases with altitude and the decreasing temperature, which affects chemical rate constants and, therefore, the relevance of specific reactions (e.g., $NO_2$ + OH).

- Line 67, "Another indicator is the ratio…": I recommend starting a new paragraph to distinguish between metrics that are used in the upper troposphere from ones that are air quality specific. Overall, this is a long paragraph that lists different metrics, and reorganizing the discussion would be helpful to communicate what's relevant for the upper troposphere and why.

The idea of this section is to discuss metrics previously reported in literature, one by one. We have tried to clarify this paragraph by starting a new line for each metric. Generally, none of these metrics is well suited to use for the upper troposphere. They have all been developed for the surface and most of them have not been applied to higher altitudes in the troposphere. The only metric that we found to be applicable for high altitudes is the HCHO to $NO_2$ ratio, but only in combination with ambient NO mixing ratios as we show later in the manuscript (e.g. Figure 6). So far, absolute values (e.g. lower than 1 for VOC sensitivity or higher than 2 for $NO_x$ sensitivity) have been reported and applied in literature. Due to the strong vertical gradients of the trace gases, these are not applicable in the upper troposphere. We have added text for clarification. Additionally, we have summarized the discussed metrics in Table 1 and we hope this provides some additional clarification for the reader.

Lines 106 ff.: These metrics were developed for the conditions near the surface, and most of them have not been applied to higher altitudes in the troposphere. Of the above metrics, the HCHO to $NO_2$ ratio is the only one, which we find to be applicable also to high altitudes – but only in combination with ambient NO mixing ratios. We demonstrate this in Section 3.3. So far, absolute thresholds (e.g. lower than 1 for VOC sensitivity or higher than 2 for $NO_x$ sensitivity (Duncan et al., 2010)) have been reported and applied in the literature. Due to the strong vertical trace gas gradients, these absolute thresholds are not applicable in the upper troposphere.

**Table 1.** Overview of the most common metrics to determine if $O_3$ chemistry is sensitive towards $NO_x$ or VOC reported in literature. Details can be found in the text.

| metric | required parameters | NO$_x$ sensitivity | VOC sensitivity | references | suitability |
|---|---|---|---|---|---|
| P(O$_3$) | NO$_x$, O$_3$, HO$_2$, RO$_2$ | P(O$_3$) increases with NO$_x$ | P(O$_3$) decreases with NO$_x$ | Brasseur et al. (1996) Jaeglé et al. (1999) Tonnesen and Dennis (2000a) Tadic et al. (2021) | surface |
| OPE | NO$_x$ & reaction products (e.g. HNO$_3$, PAN) | high OPEs | low OPEs | Liu et al. (1987) Trainer et al. (1993) Wang et al. (2018a) | surface |
| weekend O$_3$ effect | NO$_x$, O$_3$ | O$_3$ decreases on weekends | O$_3$ increases on weekends | Fujita et al. (2003) Pusede and Cohen (2012) Nussbaumer and Cohen (2020) Sicard et al. (2020) Gough and Anderson (2022) | surface |
| HCHO to NO$_2$ ratio | HCHO, NO$_2$ | high HCHO/NO$_2$ e.g. HCHO/NO$_2$ > 2* | low HCHO/NO$_2$ e.g. HCHO/NO$_2$ < 1* | Sillman (1995) *Duncan et al. (2010) Jin et al. (2020) Xue et al. (2022) | surface |
| H$_2$O$_2$ to HNO$_3$ ratio | H$_2$O$_2$, HNO$_3$ | high H$_2$O$_2$/HNO$_3$ e.g. H$_2$O$_2$/HNO$_3$ > 0.4** | low H$_2$O$_2$/HNO$_3$ e.g. H$_2$O$_2$/HNO$_3$ < 0.4** | **Sillman (1995) Wang et al. (2018b) Vermeuel et al. (2019) Liu et al. (2021) Gough and Anderson (2022) | surface |
| α(CH$_3$O$_2$) | NO, OH, HO$_2$ | α(CH$_3$O$_2$) increases with NO | α(CH$_3$O$_2$) unaffected by NO | Nussbaumer et al. (2021b) Nussbaumer et al. (2022) | entire troposphere |

- Line 95, "Khodayari et al…": what region did this modeling study focus on?

  Khodayari et al. (2018) have investigated the correlation between lightning NO$_x$ and O$_3$ production on a global scale. We have clarified this in the text.

  Line 120 f: (…) where globally a decreasing O$_3$ burden was observed with increasing lightning NO$_x$.

- Paragraph beginning line 88: This is a list of past studies that explored ozone production sensitivity regimes in the upper troposphere. In the results or conclusions, please discuss your findings with respect to these studies.

  These studies have explored ozone sensitivity based on metrics, which were developed for the surface. We expect that most of the studies published 20-25 years ago overestimated the rate constant for the reaction of NO$_2$ with OH to form HNO$_3$ (state of knowledge two decades ago), and therefore overestimate the NO$_x$ loss. We have added some discussion to Section 3.3.2 and also highlight this aspect.

  Lines 467 ff.: Our results fit well with previous literature on the role of lightning NO$_x$ in O$_3$ production in the upper troposphere. Grewe (2007) reported that NO$_x$ from lightning is the dominant source of O$_3$ in the upper tropical troposphere based on simulations with the global climate–chemistry model E39/C. Similar results have been presented by Murray (2016), Schumann and Huntrieser (2007) and Sauvage et al. (2007). While it has been shown previously that ozone production in the UTT is highly dependent on lightning NO$_x$, this study is the first to extensively study the impact of lightning on the dominant O$_3$ regime, applying a new indicator α(CH$_3$O$_2$), which is valid throughout the entire troposphere.

- Figure 1: Is this from the EMAC modeling output, described in section 2.3? If so, state in the figure caption and description of the figure.

The black curve and grey error shade represent an average of all available data points at 200hPa in the tropics for $\alpha(CH_3O_2)$ binned to NO from the EMAC modeling output. The red line presents a linear fit of the data points for NO < 0.1ppbv and the blue line a linear fit for NO > 0.1ppbv. We have added some details in the figure caption and the text.

**Figure 1**. Illustration of how $\alpha(CH_3O_2)$ can serve as a metric to identify the dominant chemical regime. The black line shows the tropical background $\alpha(CH_3O_2)$ binned to NO mixing ratios from the EMAC model output. The grey shading shows the 1$\sigma$ standard deviation. The red dashed line is a linear fit for NO < 0.1 ppbv and represents $NO_x$-sensitive $O_3$ chemistry. The blue dashed line represents VOC-sensitive $O_3$ chemistry for NO > 0.1 ppbv.

and

Line 177 f.: (…) as obtained from the EMAC modeling output.

- Supplemental Figure S1: Say in the figure caption what the black line and grey shading are. Are these mixing ratios all sampled at local noon?

The black line shows the yearly averages of daily values at local noon of each respective species. The grey shades represent the variability, as 1$\sigma$ standard deviation of the averaging. We have added this information in the figure caption.

**Figure S1.** Development of modeled trace gas mixing ratios in the upper tropical troposphere at 200 hPa from 2000 to 2019. Black lines represent the average of daily mixing ratios at local noon for each year, and the grey shades show the variability, calculated as the 1$\sigma$ standard deviation of the averaging.

- Lines 194 – 204: This text is hard to follow. If the authors find these individual mixing ratios are useful, perhaps they should be given as a table or separate figure. If not, could they be summarized to support the support the discussion of the patterns in NO that are due to seasonal and continental vs marine features? Throughout the paper there is text where the authors list values for multiple locations. If there are common features of these locations (i.e., Southern vs Northern Hemisphere) that are driving the patterns seen in the mixing ratio or sensitivity regimes, please point this out instead.

We regret that the referee finds this section not clear enough. However, we think it is important to provide a description of the figure, including absolute values, to the reader prior to the interpretation of the results. We have restructured this paragraph and focus on relative comparisons between areas, including maritime and continental areas, as well as northern and southern hemispheric values, and provide absolute average values in brackets instead. We hope this helps to improve the clarity for the reader.

Lines 238 ff.: During DJF, NO mixing ratios are highest over the tropical continental areas of South America, southern Africa and northern Australia, with average peak values between 0.3 and 0.4 ppbv. In comparison, NO mixing ratios are much lower

over the Pacific and the Indian Oceans (0.09 ± 0.01 ppbv), the Atlantic Ocean (0.12 ± 0.02 ppbv), North Africa (0.10 ± 0.03 ppbv) and South East Asia (0.08 ± 0.01 ppbv). Generally, the mixing ratios over land are much higher than those over the ocean, and the mixing ratios north of the equator (0.09 ± 0.03 ppbv) are lower compared to south of the equator (0.14 ± 0.06 ppbv). During MAM, NO mixing ratios over South America are similar to those during DJF (0.21 ± 0.05 ppbv). NO mixing ratios over Africa are almost twice as high on average compared to DJF and reach peak values of 0.53 ppbv. The relative maxima relocate from South to Central Africa from DJF to MAM and also over the Arabian Peninsula and South Asia, including India. Mixing ratios over Australia are around 0.15 ppbv and approximately half of those during DJF and are similar over South East Asia. During JJA, peak NO mixing ratios are found north of the equator over Central America and North Africa. Average NO mixing ratios are almost 60 % higher in the northern (0.14 ± 0.07 ppbv) compared to the southern tropical hemisphere (0.09 ± 0.02 ppbv). The distribution, therefore, differs drastically from DJF. During SON, NO mixing ratios are similar to MAM and peak over South America and Central Africa.

We performed according adjustments to the remaining sections.

- Line 209, "During DJF…": This should be stated earlier in the discussion, with citation.

We have added this information in the introduction of Section 3.2.1.

Lines 232 ff.: We find large changes throughout the year related to the seasonality of deep convection and the location of the intertropical convergence zone (ITCZ). Deep convection dominates in the southern hemisphere in January, and during July it is most prevalent in the northern hemisphere Yan (2005).

- Line 240: What specifically about your modeled NOPR values is in line with the findings by Apel et al. (2015)?

Apel et al. (2015) observed enhanced $O_3$ production for high $LNO_x$, which fits well with our results that NOPR and NO mixing ratios show the same features considering their geographic distribution. We have clarified this in the text.

Lines 283 ff.: These results, showing similar features for NOPRs and NO geographically, are in line with findings by Apel et al. (2015), who reported enhanced ozone production for high lightning NOx over the U.S. during a research flight in June 2012 as part of the DC3 campaign.

- Line 273: What do you mean by "mostly absolute values" here; is this columns of HCHO and $NO_2$?

With absolute values we mean that in order to determine the dominant sensitivity using the HCHO/$NO_2$ ratio, in literature absolute thresholds are compared. The ratio is calculated and the resulting number is compared to a threshold, e.g. 1 for a VOC sensitivity or 2 for a $NO_x$ sensitivity (according to Duncan et al., 2010). However, these thresholds (represented by absolute numbers) can vary by location

and have never been explored for the free or upper troposphere. We have clarified this in the text.

Lines 317 ff.: In the literature, mostly absolute values for the HCHO/NO$_2$ ratio are used to determine the chemical regime, meaning the ratio is calculated and compared to a certain threshold, for example HCHO/NO2 > 2 for NOx sensitivity and HCHO/NO2 < 1 for VOC sensitivity (Duncan et al., 2010).

- Lines 289 – 318: Please consider more concise wording for the description of Figure 6. What are the main results you want the reader to understand from this Figure?

We have added some text to improve the structure of this section and hope it provides more clarity.

Lines 303 ff.: In this section, we first discuss the background curves (representing an average of all data point), and we subsequently present our findings for the individual areas.

Lines 335 ff.: To investigate individual areas regarding predominant O$_3$ sensitivity, we analyze the location of the area averages along the background. Figure 6(a) shows NO vs α(CH$_3$O$_2$) during DJF. (…)

Lines 370 ff.: In summary, Figure 6 illustrates three important results. First, the transition between NO$_x$ and VOC sensitivity occurs at around 0.1 ppbv NO in the upper tropical troposphere. Second, areas with increased lightning activity tend towards the dominance of VOC sensitivity. And third, the dominating regime changes with the time of the year. We will discuss these findings and their implications in the following.

- Figure 7: Add the season (MAM) to the figure caption.

We have added MAM to the figure caption.

**Figure 7.** Map of the tropical UT between 30°S and 30°N colored by the slopes of NO vs α(CH$_3$O$_2$) of the data in model grid regions, exemplary for MAM. (…)

- Line 345: Is the 38 ppbv$^{-1}$ value for the slope?

Yes, this value describes the slope. We have added this information.

Line 396: For the free troposphere, we find strong NO$_x$ sensitivity with a maximum slope of 38 ppbv$^{-1}$.

- Figure 9: If the average backgrounds (or another metric) from the base and other sensitivity studies are shown here, the readers can more easily compare these results to those shown in Figure 6 and S16-S17. Additionally, it might be helpful to move Figure S18 to the main text since the results of the aircraft emission sensitivity study is not currently shown in the main text figures.

Thank you for these suggestions. We have added a comparison of the baseline scenario and the scenario without lightning to Figure S15 of the Supplement. Additionally, we have moved Figure S18 to Figure 10 of the main text in order to show both sensitivity studies (for lightning and aircraft).

Lines 414 f.: Figure S14 of the Supplement shows the subdivision into the four periods (DJF, MAM, JJA and SON), and in Figure S15, we present a comparison between the baseline scenario as a yearly average and the scenario excluding lightning.

**Figure S15**. Comparison of the plots shown in Figure 6 ($\alpha(CH_3O_2)$, $O_3$ and the HCHO/$NO_2$ ratio binned to NO mixing ratios) for the modeling scenario with (bottom row) and without lightning (top row). The baseline scenario is here shown as a yearly average. Black lines show averages of all data points, and grey shades present the 1$\sigma$ standard deviation. Colored data points show the averages for the indicated areas with the 1$\sigma$ error.

[Figure]

- Line 378: What do you mean by "the absolute values" here? Is this referring to the ratio for individual regions?

We are trying to say that the course of the black line is similar for both scenarios (baseline and excluding lightning), but the values for the HCHO/$NO_2$ ratio are notably higher for the scenario where lightning was excluded. We have rephrased this for clarification.

Line 429 f.: The course of the average values (black line) is again similar to the one for the baseline scenario, but the values for the HCHO/$NO_2$ ratio are higher.

- Lines 406 – 412: How do these results fit in with past studies?

We have added some text to account for past studies.

Lines 467 ff.: Our results fit well with previous literature on the role of lightning $NO_x$ in $O_3$ production in the upper troposphere. Grewe (2007) reported that $NO_x$ from lightning is the dominant source of $O_3$ in the upper tropical troposphere based on simulations with the global climate–chemistry model E39/C. Similar results have been presented by Murray (2016), Schumann and Huntrieser (2007) and Sauvage et al. (2007). While it has been shown previously that ozone production in the UTT is highly dependent on lightning $NO_x$, this study is the first to extensively study the impact of lightning on the dominant $O_3$ regime, applying a new indicator $\alpha(CH_3O_2)$, which is valid throughout the entire troposphere.

- Line 417: Was an additional run without aircraft emissions conducted?

Yes, a model run without aircraft emissions was performed. We have added this in the conclusion section.

Lines 477 ff.: One model run considers a baseline scenario and four additional ones were run with halved, doubled and excluded lightning $NO_x$ emissions, as well as excluded $NO_x$ aircraft emissions.

- Lines 437- 441: Citations should be provided for these statements.

We have added several references indicating that $NO_2$ measurements in the upper troposphere suffer from drawbacks e.g. thermal reservoir species (methyl peroxy nitrate) which release $NO_2$ in instrumental parts with elevated temperatures.

Lines 502 f.: $NO_2$ measurements tend to suffer from the unselective detection or artifacts from reservoir species, which makes accurate quantification challenging (Reed et al., 2016; Jordan et al., 2020; Andersen et al., 2021; Nussbaumer et al., 2021b)

- Conclusion section: The discussion of the role of lightning in the conclusions is terse and doesn't explain how the study reached these results or how it fits in with the current literature.

We mostly discuss the role of lightning in Section 3.3.2 as part of the sensitivity study. We have added some extra discussion, particularly in regard to the literature (see above).

Lines 467 ff.: Our results fit well with previous literature on the role of lightning $NO_x$ in $O_3$ production in the upper troposphere. Grewe (2007) reported that $NO_x$ from lightning is the dominant source of $O_3$ in the upper tropical troposphere based on simulations with the global climate–chemistry model E39/C. Similar results have been presented by Murray (2016), Schumann and Huntrieser (2007) and Sauvage et al. (2007). While it has been shown previously that ozone production in the UTT is highly dependent on lightning $NO_x$, this study is the first to extensively study the impact of lightning on the dominant $O_3$ regime, applying a new indicator $\alpha(CH_3O_2)$, which is valid throughout the entire troposphere.